# G-quadruplexes promote the motility in MAZ phase-separated condensates to activate CCND1 expression and contribute to hepatocarcinogenesis

Wenmeng Wang [1,5], Dangdang Li [1,5] ✉, Qingqing Xu[1], Jiahui Cheng [1], Zhiwei Yu[2], Guangyue Li [1], Shiyao Qiao[1], Jiasong Pan[1], Hao Wang[1], Jinming Shi[1], Tongsen Zheng[3,4] & Guangchao Sui [1] ✉

G-quadruplexes (G4s) can recruit transcription factors to activate gene expression, but detailed mechanisms remain enigmatic. Here, we demonstrate that G4s in the CCND1 promoter propel the motility in MAZ phase-separated condensates and subsequently activate CCND1 transcription. Zinc finger (ZF) 2 of MAZ is a responsible for G4 binding, while ZF3-5, but not a highly disordered region, is critical for MAZ condensation. MAZ nuclear puncta overlaps with signals of G4s and various coactivators including BRD4, MED1, CDK9 and active RNA polymerase II, as well as gene activation histone markers. MAZ mutants lacking either G4 binding or phase separation ability did not form nuclear puncta, and showed deficiencies in promoting hepatocellular carcinoma cell proliferation and xenograft tumor formation. Overall, we unveiled that G4s recruit MAZ to the CCND1 promoter and facilitate the motility in MAZ condensates that compartmentalize coactivators to activate CCND1 expression and subsequently exacerbate hepatocarcinogenesis.

Cancer is characteristic of uncontrollable cell proliferation governed by deregulated gene expression, which is modulated by various transcription factors (TFs), such as p53 and MYC[1,2]. TFs activate gene transcription through binding to target promoters and, in many cases, enhancers[3]. In this process, a TF decodes double-stranded DNA (dsDNA) sequences in the genome to recognize and bind its consensus binding sites and recruit transcriptional coactivators (such as MED1 and BRD4), transcription initiation and elongation factors (such as CDK7 and CDK9), and RNA polymerase II (Pol II) to initiate transcription[3,4]. Therefore, specific DNA sequences and their inherent structural motifs are key chromatin elements to recruit TFs for subsequent gene expression.

Unlike regular double-helix DNA structure, G-quadruplexes (G4s) represent a type of non-canonical and stable nucleic acid secondary structures, and are mostly found in guanine-rich sequences with at least four adjacent consecutive G-runs or G-tracts. G4 structures are built of stacked G-quartets constituted of four Hoogsteen hydrogen-bonded guanine bases[5,6]. In the past two decades, G4s have attracted increasing attentions in the cancer research field. G4s' formation and resolving are closely associated with diverse biological processes, especially cancer-related gene expression, and G4 motifs have been considered as promising targets in cancer therapies[5,6]. Noteworthily, recent studies revealed a mechanism of G4-mediated TF recruitment, implying promoter G4s as regulatory hubs to recruit TFs and activate transcription[7,8]. Importantly, G4s bind TFs more effectively than regular dsDNA does[7,9], and many TFs are significantly enriched at genomic G4 sites in cancer cells[7,8]. In addition, mounting evidence suggests that

[1]College of Life Science, Northeast Forestry University, Harbin 150040, China. [2]Department of Colorectal Surgery, Harbin Medical University Cancer Hospital, Harbin 150081, China. [3]Department of Gastrointestinal Medical Oncology, Harbin Medical University Cancer Hospital, Harbin 150081, China. [4]Key Laboratory of Molecular Oncology of Heilongjiang Province, Harbin, China. [5]These authors contributed equally: Wenmeng Wang, Dangdang Li. ✉e-mail: lidd@nefu.edu.cn; gcsui@nefu.edu.cn

G4 enrichment in the genome was associated with increased potential of oncogenesis[8,10–12], and consistently, G4 structures were significantly elevated in cancerous cells and tissues[8,10,13]. Moreover, G4s are prevalently discovered in the promoters of frequently amplified and/or highly expressed oncogenes[11]. Recently, promoter G4s were shown to recruit Pol II and activate gene transcription in cancer cells[12]. As such, G4s may also act as transcriptionally active elements to drive oncogenes. However, detailed mechanisms underlying G4-activated transcription through TF and coactivator enrichment, and subsequently assembling transcription complexes have remained mysterious.

Liquid-liquid phase separation (LLPS) has emerged as an increasingly recognized mechanism depicting the compartmentalization of biomolecules, including proteins, DNAs and RNAs, into condensates involved in a variety of cellular processes[14,15]. The weak multivalent interactions and/or direct binding among different proteins, or proteins and nucleic acids, play a crucial role in the formation of cellular LLPS condensates[15,16]. For instance, dsDNA binding to cyclic GMP-AMP synthase (cGAS) induces LLPS of DNA-cGAS condensation that triggers innate immune signaling[17]. Additionally, dsDNA harboring TF binding sites drives condensate formation of TFs and coactivators[18]. These DNAs and RNAs can serve as scaffolds for the assembly of phase-separated condensates[19–22]. Of note, both the MYC promoter and telomeric G4 motifs displayed strong ability to increase LLPS of histone H1 in vitro. Only G4s with parallel topology could bind SERBP1, an RNA-binding protein, to promote its nuclear phase-separated condensation[23,24]. The sequences of many cancer-related gene promoters or enhancers may form different G4 conformations, including parallel, antiparallel and hybrid[5]. To date, it is still unclear whether these G4 motifs are causally related to the phase-separated condensation of TFs.

The MYC-associated zinc finger (MAZ) protein was previously reported to bind G4s[25–27]. Importantly, as a TF, MAZ has six $C_2H_2$ zinc fingers that specifically recognize a G/C-rich motif[28]. MAZ are overexpressed in cancers and promotes cancer development and metastasis[29]. However, to date, limited research has been reported for MAZ-related regulatory mechanisms, especially in hepatocellular carcinoma (HCC). Intriguingly, a gene activation model through phase-separated condensate formation by TFs and their coactivators was proposed[14,30]. Given that MAZ is a G4-binding protein with oncogenic activity, we hypothesize that its G4 binding ability in oncogene promoters is causally associated with MAZ phase separation, leading to the formation of MAZ/G4 phase-separated condensates that activates oncogene expression.

As a proliferative gene, CCND1 (or cyclin D1) plays an essential role in cell cycle progression and is a potential therapeutic target[31]. The human CCND1 gene promoter has a high G/C content, especially in the region close to the transcription start site (TSS), while MAZ's consensus binding sites GGG(A/C)GGG are G/C-rich[28]. Thus, we verified MAZ binding to the CCND1 promoter and took MAZ-mediated CCND1 activation as an example in this study. Using a suit of approaches, we determined G4 formation by a G-rich stretch in the CCND1 promoter in vitro and in cells. Importantly, we observed that the G4s could recruit MAZ, promote MAZ/G4 co-condensation, and facilitate the motility of MAZ condensates that incorporates many transcription coactivators to activate CCND1 expression and accelerate oncogenic phenotypes of HCC. Overall, this study demonstrates the functions of G4s in recruiting TFs and other transcriptional regulators, and promoting their LLPS to activate oncogene expression.

## Results

### The CCND1 promoter folds into G4s in vitro and in cells
The human CCND1 promoter has multiple consecutive G-tracts in the negative strand (Supplementary Fig. 1a), indicating G4 forming potential. Consistently, previous ChIP-seq studies using a G4 antibody, BG4[5,10], demonstrated G4 enrichment in the CCND1 gene locus with higher signal than many other G4-containing oncogenes[10], suggesting G4 formation in its promoter.

High G/C sequence resides between −500 and −1 of the CCND1 promoter, with the TSS as +1 (Supplementary Fig. 1a). QGRS Mapper[32] analysis revealed four consecutive 5G-tracts between −134 and −105 on the negative strand with a G4-score higher than reported MYC promoter G4 (MYC-G4) (Supplementary Fig. 1a, b). Multiple 3G- and 5G-tracts are mingled together, which is evolutionarily conserved in the CCND1 promoters of eight species from rodents to primates (Supplementary Fig. 1c). Since these adjacent G-tracts may fold into various G4s in different combinations, we hereafter focused on −142 to −92 sequence covering all eight 3 G/5G-tracts. In synthetic wild type (WT) and mutant (Mut1 and Mut2) oligonucleotides (oligos), Mut1 had all G-tracts disrupted by G-to-A substitutions, and Mut2 retained separated G-tracts to prevent intramolecular G4 formation (Fig. 1a). MYC-G4 was used as a positive control.

In CD spectrometry to evaluate in vitro G4 formation, both WT and MYC-G4 oligos annealed in the presence of KCl displayed positive and negative molar ellipticity peaks at 262 and 240 nm, respectively, indicating parallel G4 topology (Supplementary Fig. 1d). In LiCl or no cation solution, these peaks were substantially reduced. Mut2 annealed in KCl solution also displayed these characteristic peaks, but with markedly decreased molar ellipticity, especially at 262 nm. Mut1 exhibited peaks at 245 nm and 270 nm with or without cation, suggesting structures different from G4s (Supplementary Fig. 1d). Furthermore, WT and Mut2 showed comparably strong thermostability to MYC-G4 in KCl solution, but Mut1's structures quickly disassembled with temperature increase under all conditions (Supplementary Fig. 1e). Overall, these results suggest potential parallel G4 structure formation by CCND1 promoter G-tracts.

In EMSA using FAM-labeled oligos, BG4, but not a control antibody, strongly bound WT oligo, especially in KCl solution. BG4 showed reduced binding to Mut2, but lost affinity to Mut1 (Fig. 1b), in line with their G4 forming ability. Consistently, FAM-labeled WT and BG4 complex was competitively disrupted by unlabeled WT and Mut2 oligos, but not Mut1, although Mut2 showed less ability (Supplementary Fig. 2a).

Interestingly, BG4 binding to WT oligo significantly declined under conditions of increased TMPyP4 or PDS, two well-characterized G4 stabilizers[10,33], suggesting that their incorporation into G4s blocked BG4 binding (Supplementary Fig. 2b). Importantly, both WT and Mut2, but not Mut1, showed sparkles in HeLa cells immunostained by BG4, suggesting their G4 formation in a cellular environment (Fig. 1c). In ChIP-PCR assays of HepG2 and SMMC-7721 cells, the CCND1 promoter G4 motif region was immunoprecipitated by BG4 but not a control IgG, corroborating its G4 formation ability (Fig. 1d).

The Hoogsteen hydrogen bonds in G-quartets protect the guanines from DMS-mediated methylation, and subsequently prevent their cleavage by piperidine[34]. Based on relative intensity of guanine bands in DMS footprinting assays, the five G-tracts of WT oligo (red letters in Fig. 1e) annealed in KCl solution were primarily engaged in G4s. Notably, Mut2 annealed in KCl solution showed only six guanines ($G_{-107}$-$G_{-109}$ and $G_{-111}$-$G_{-113}$) involved in G4 formation (Supplementary Fig. 2c, right), suggesting its intermolecular G4 formation, mostly in parallel topology as revealed in Supplementary Fig. 1d. In contrast, Mut1 displayed no discernable difference of guanine signals under different conditions (Supplementary Fig. 2c, left), implying its lack of G4 forming ability. Thus, WT and Mut2 oligos could form intramolecular and/or intermolecular G4s in potassium solution (Fig. 1f). Collectively, our data strongly supported G4 formation in the CCND1 promoter.

### G4s in the CCND1 promoter act as transcriptional activator elements
To test whether G4s regulate the CCND1 promoter, we generated a Gluc reporter construct, pCCND1-WT-Gluc (WT), and its three G4-

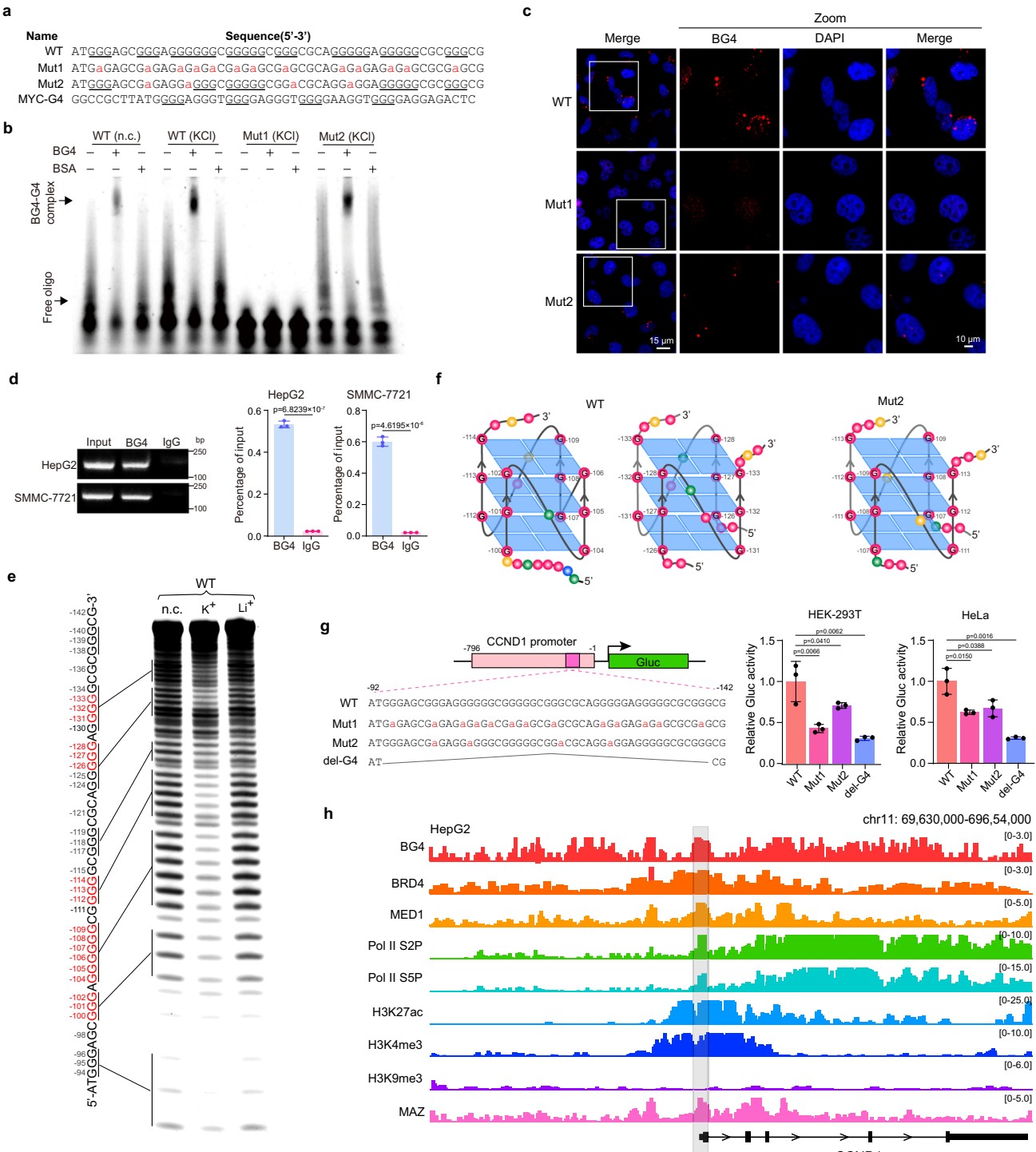

**Fig. 1 | Determination of G4 formation in the CCND1 promoter and its relevance to CCND1 gene activation. a** Sequences of synthetic oligonucleotides (oligos) based on the negative strand of the CCND1 promoter. G-tracts are underlined and G-to-A mutations are shown in red and small letters. WT, wild-type; Mut1 (mutant 1): all G-tracts disrupted; Mut2 (mutant 2): four indicated G-tracts disrupted. MYC-G4 (from the human MYC promoter) was used as a positive control. **b** EMSA analysis of BG4 binding to FAM-labeled oligos. **c** Immunofluorescence (IF) studies using BG4 to detect G4s formed by oligos transfected into HeLa cells. Scale bars: left panel: 15 μm; right panels: 10 μm. **d** ChIP studies of BG4 binding to the CCND1 promoter in HCC cells. Normal IgG was used as a control antibody. **e** DMS footprinting assays to identify guanines involved in G4 formation in the CCND1 promoter. DNA sequences corresponding to the CCND1 promoter were labeled. Red letters denote protected guanines. **f** Schematic models of CCND1-G4 structures formed by WT (left and

middle) and Mut2 (right) oligos. WT oligo can form intramolecular parallel G4s using four 3G-tracts (G$_{-100}$-G$_{-102}$, G$_{-104}$-G$_{-106}$, G$_{-107}$-G$_{-109}$ and G$_{-112}$-G$_{-114}$) with one one-nucleotide, one zero-nucleotide and one two-nucleotide loops (left). WT and Mut2 may form intermolecular parallel G4s using two 3G-tracts (WT: G$_{-126}$-G$_{-128}$ and G$_{-131}$-G$_{-133}$; Mut2: G$_{-107}$-G$_{-109}$ and G$_{-111}$-G$_{-113}$) (middle and right). **g** Reporter assays to determine the effects of G4s on CCND1 promoter activity. Left, representation of WT and mutant CCND1 promoter reporter constructs with mutated guanines in red. **h** ENCODE epigenetic information across the CCND1 gene locus in HepG2 cells (chr11:69,630,000-696,54,000 [hg19]). ChIP peaks located in the CCND1 gene promoter region is shaded in grey. In (**b**, **c**, **e**), data are representative of three biologically independent experiments. In (**d**, **g**), data are mean ± s.d. (*n* = 3 biologically independent experiments). Unpaired two-tailed Student's *t*-test was used for statistical analysis. Source data are provided as a Source Data file.

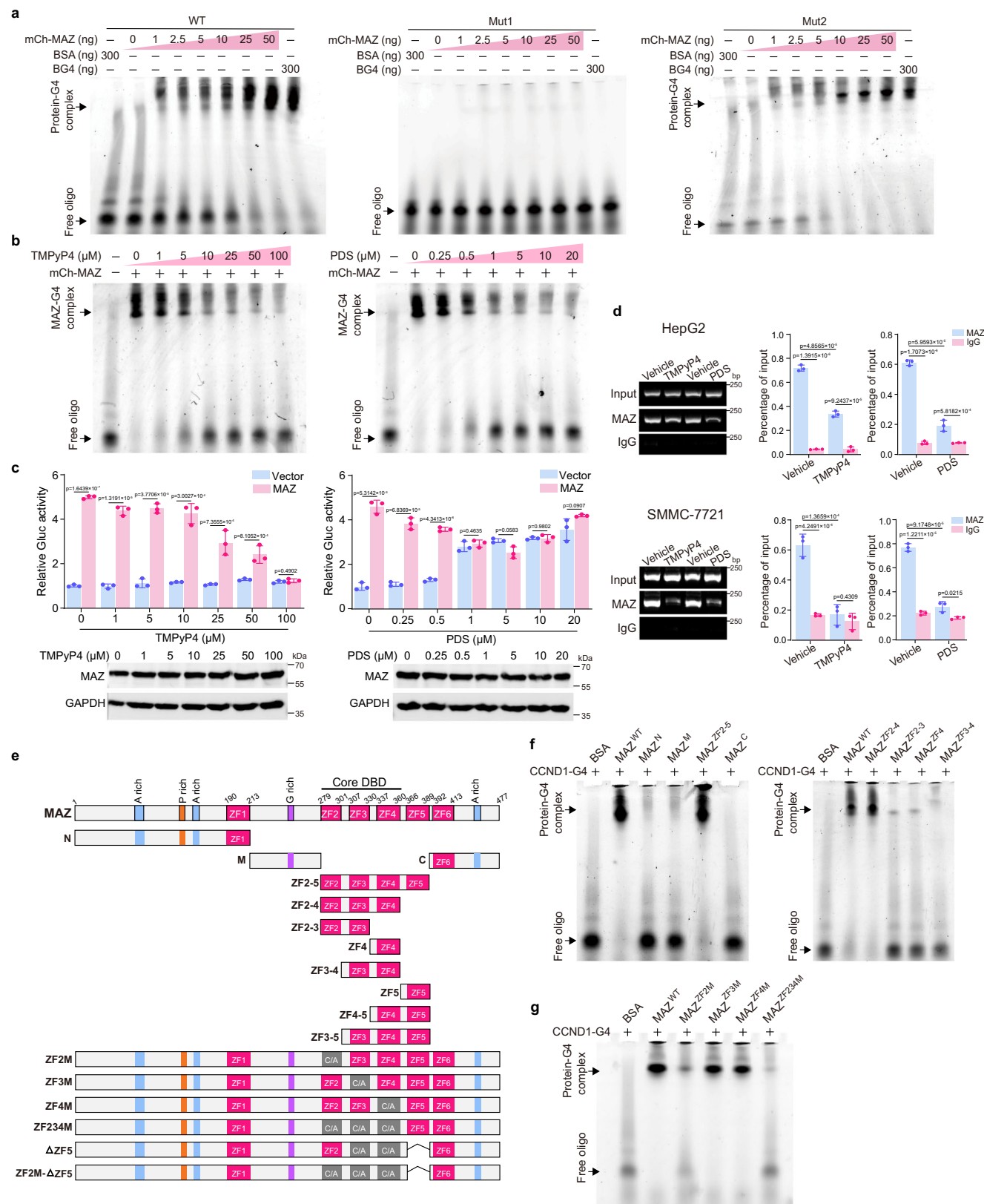

mutated or -deleted reporters (Mut1, Mut2 and del-G4) (Fig. 1g, left). In reporter assays, all three G4-mutated reporters showed significantly decreased Gluc activity versus WT reporter in HEK-293T and HeLa cells (Fig. 1g, middle and right), suggesting G4s' positive role in CCND1 promoter activity.

Next, integrative ChIP-seq data analyses of BG4 and multiple transcriptional coregulators in HepG2 cells using ENCODE datasets revealed overlapped enrichment between G4s and active transcription markers, including BRD4, MED1, RNA Pol II, H3K27ac and H3K4me3, but not a repressive histone marker H3K9me3, in the CCND1 promoter

**Fig. 2 | MAZ interacts with CCND1-G4 and activates its transcription. a, b** EMSA analyses of MAZ binding to oligos derived from the CCND1 promoter. Increasing amounts of purified recombinant MAZ were incubated with FAM-labelled WT, Mut1 and Mut2 oligos annealed in KCl-containing solution. BSA (bovine serum albumin) and BG4 were used as negative and positive controls, respectively. In (**b**), FAM-labelled WT oligo annealed in KCl-containing solution was mixed with increasing amounts of TMPyP4 and PDS, followed by the addition of MAZ. **c** Reporter assays to examine effects of TMPyP4 and PDS on MAZ-regulated CCND1 promoter activity. CCND1 promoter reporter (WT) and pSL4-3×Flag-MAZ plasmids were cotransfected with pCMV-SEAP into HeLa cells, followed by the treatment of increasing concentrations of TMPyP4 and PDS. Relative Gluc activity was presented after normalized against SEAP activity. **d** ChIP studies of the effects of TMPyP4 and PDS on MAZ binding to the CCND1 promoter. HepG2 and SMMC-7721 cells were treated by 25 μM TMPyP4 or 20 μM PDS followed by ChIP assays using an MAZ antibody and control IgG. **e** Domain structures of MAZ WT and its mutants. In the last six mutants, C/A denotes the C-to-A mutation in each ZF. **f, g** EMSA analyses of CCND1-G4 interactions with MAZ WT and indicated mutants. In (**a, b, f, g**), data are representative of three biologically independent experiments. In (**c, d**), data are mean ± s.d. (*n* = 3 biologically independent experiments). Unpaired two-tailed Student's *t*-test was used for statistical analysis. Source data are provided as a Source Data file.

(Fig. 1h). Thus, bioinformatic analyses support G4s as positive elements for CCND1 promoter transcription. Consistently, immunofluorescence assays in HepG2 and SMMC-7721 cells clearly demonstrated colocalization of BG4 with active transcription markers, including BRD4, MED1, CDK9, Pol II S2P/S5P, H3K27ac and H3K4me3, but not H3K9me3 (Supplementary Fig. 3a–d). Together, our data strongly support that G4s activate CCND1 gene expression in HCC cells.

## MAZ interacts with G4s of the CCND1 promoter and activates its transcription

Multiple TFs with G/C-rich binding motifs bind G4s[9,26,35]. In the CCND1 promoter, the JASPAR software[36] identified six putative MAZ-binding sites, designated as BS1 to BS6 (Supplementary Fig. 4a). MAZ binding to this region, especially BS1, BS2 and BS3, was verified in EMSAs using FAM-labeled oligos and purified His×6-MAZ (Supplementary Fig. 4b–e).

Since MAZ binds G4s in the KRAS, HRAS and MYB promoters[25–27], we tested its binding to CCND1-G4s in EMSA and observed MAZ's dose-dependent interaction with WT and Mut2 oligos, but not Mut1 (Fig. 2a). Consistently, MAZ and FAM-labeled WT complex was disrupted by unlabeled WT and Mut2, but not Mut1 (Supplementary Fig. 4f). Additionally, both TMPyP4 and PDS reduced MAZ-WT oligo interaction (Fig. 2b), reminiscent of their adverse effects on BG4-G4 binding (Supplementary Fig. 2b), suggesting that MAZ did not well recognize TMPyP4-/PDS-associated G4s. Interestingly, MAZ formed complexes with previously reported G4s in the MYC, BCL2, MYB, MDM2 promoters and telomere (TEL) in a dose-dependent manner, which was verified in oligo competition experiments (Supplementary Fig. 5a–e). The data suggest MAZ as a genuine G4-binding protein.

Next, MAZ activated the CCND1-promoter reporter, which was dampened by TMPyP4 and PDS (Fig. 2c), consistent with their adverse effects on MAZ-G4 interaction. In this study, gradually increased reporter activity at high concentrations of PDS, but not TMPyP4, was likely due to their binding to different positions in G4 structures[37,38]. Consistently, ChIP-PCR verified reduced MAZ binding to the CCND1 promoter in TMPyP4-/PDS-treated HCC cells (Fig. 2d). Together, our data support that MAZ activates CCND1 transcription through binding to its promoter G4s.

## Zinc fingers 2-4 determine MAZ interaction with G4s

MAZ protein is highly conserved in vertebrate species (Supplementary Fig. 6a). Among the six $C_2H_2$ zinc fingers (ZFs) (Fig. 2e and Supplementary Fig. 6b), ZF2-4 recognize dsDNA and are identical in these species, suggesting MAZ's functional conservation.

We generated a series of truncated MAZ mutants to test its G4-binding regions (Fig. 2e) and purified their mCherry-fusion proteins (Supplementary Fig. 7a–c). In EMSA, only MAZ(ZF2-4), but not ZF2-3, ZF4 or ZF3-4, formed complex with CCND1-G4 (Fig. 2f). As cysteines (Cs) are essential for ZF's DNA binding, we next individually replaced them by alanines to assess ZF2-4's role in MAZ-G4 interaction. Full-length MAZ mutants ZF2M and ZF234M holding C-to-A mutations in ZF2 and all three ZFs, respectively, virtually failed to form complex with CCND1-G4, while ZF3M and ZF4M largely retained this ability (Fig. 2g). The same binding pattern of these MAZ mutants to G4s in MYC, BCL2, MYB, MDM2 and TEL was observed (Supplementary Fig. 7e–i). Thus, ZF2 is an essential, but not sufficient, element for MAZ to bind CCND1-G4, while the concurrent presence of ZF2-4 is required for this binding.

## MAZ is overexpressed in liver cancer and undergoes phase separation in vitro and in cells

When examining MAZ levels in liver cancer, we observed increased MAZ expression in various HCC cell lines and 8 liver cancer samples, compared to nontumorigenic hepatic HL-7702 cells and matched normal liver tissues, respectively (Supplementary Fig. 8a–c). Consistently, analysis of a TCGA dataset[39] also demonstrated robust and highly significant MAZ increase in HCC samples versus match normal liver tissues (Supplementary Fig. 8d). Intriguingly, endogenous MAZ formed nuclear puncta in both hepatic cells and tissues, with larger numbers and sizes in cancerous cells than nontumorigenic samples (Supplementary Fig. 8e and Fig. 3a). When analyzed by PONDR[40], about 75% of MAZ sequence were intrinsically disordered regions (IDRs) (Fig. 3b), suggesting its propensity of undergoing phase separation[14,16]. Consistently, mCherry-MAZ formed droplets in vitro in the presence of 10% PEG 8000, a molecular crowding reagent (Fig. 3c). Droplet numbers and sizes accelerated with increased mCherry-MAZ concentrations and PEG molecular weights (Fig. 3c and Supplementary Fig. 9a, b), while droplet sizes, but not numbers, were significantly larger at 25 °C and 37 °C than those at 4 °C (Supplementary Fig. 9c). Importantly, formation of nuclear puncta and in vitro droplets by mCherry-MAZ dramatically reduced with increased concentrations of 1,6-hexanediol (1,6-hex), which disrupted phase-separated condensation[41,42] (Fig. 3d and Supplementary Fig. 9d).

Based on microscopic observation and solution turbidity, mCherry-MAZ droplet formation favored low salt levels, but gradually declined with NaCl or KCl increase, even within KCl's physiological concentrations (60–150 mM)[43] (Supplementary Fig. 9e–h). Most phase-separated proteins display quick fusion of adjacent condensates and rapid fluorescence recovery after photobleaching (FRAP) due to dynamic molecule movement[41,42]. Strikingly, mCherry-MAZ showed remarkably slow or incomplete fusion of adjacent droplets and lacked significant FRAP in vitro (Fig. 3e, f and Supplementary Movie 1). However, mCherry-MAZ displayed nearly complete condensate fusion and slow FRAP in cells (Fig. 3g, h and Supplementary Movie 2), implicating the presence of cryptic regulatory mechanisms driving MAZ LLPS in a cellular environment. Together, our data strongly support that MAZ undergoes phase separation both in vitro and in cells.

MAZ has four featured poly-amino acid regions, especially two A-rich clusters at the N- and C-termini (Supplementary Fig. 6b), which potentially contribute to phase separation of IDR-containing proteins[41,44]. Surprisingly, only ZF3-5-containing MAZ mutants (i.e., ZF2-5 and ZF3-5, Fig. 2e), but none of others, including the N mutant with the highest IDR score, formed droplets (Fig. 3i), suggesting that ZF3-5 are critical for MAZ phase separation. Furthermore, mCherry-

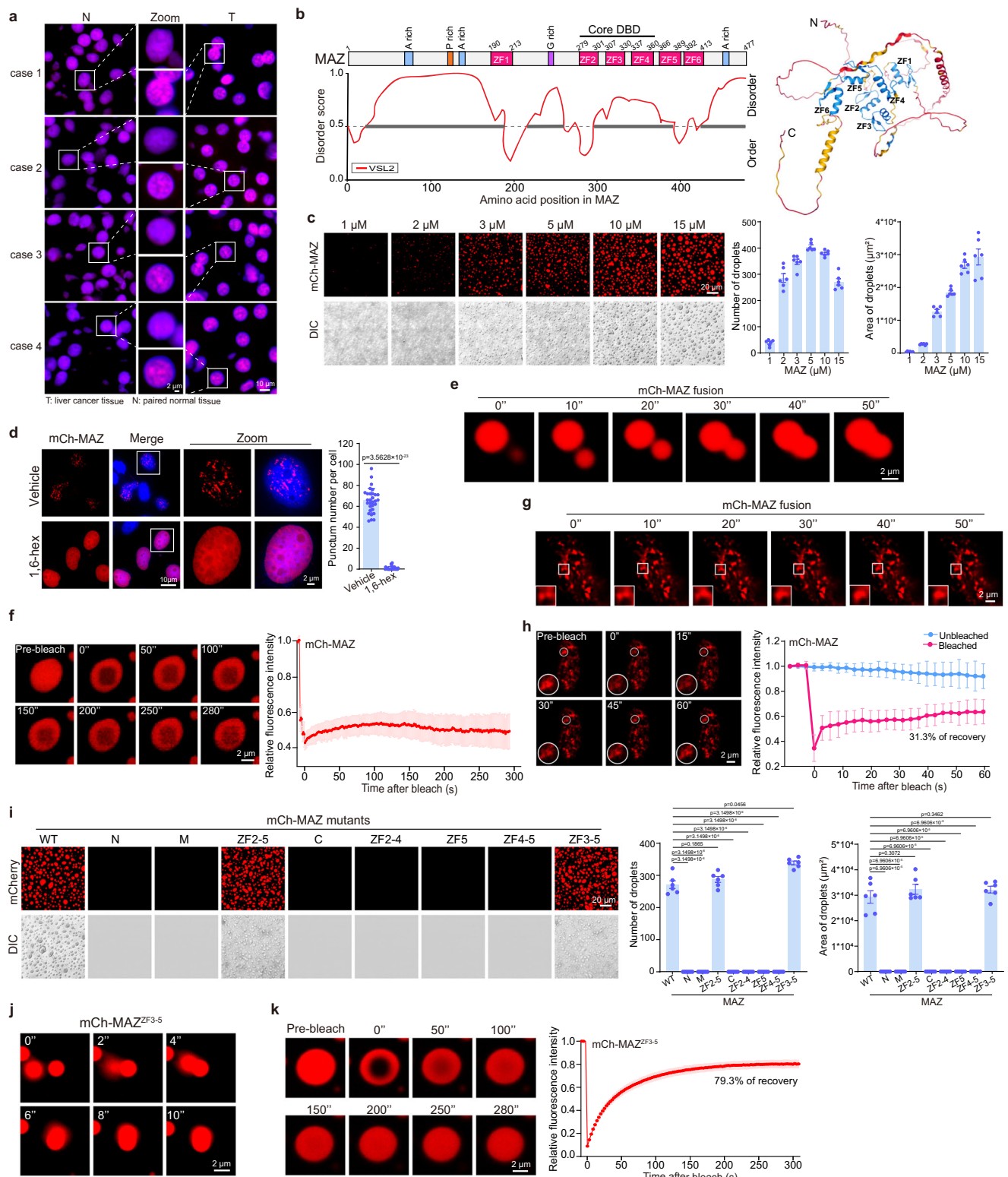

MAZ-ZF3-5 condensates displayed prompt droplet fusion and FRAP (Fig. 3j, k and Supplementary Movie 3), with the rates much faster than WT MAZ (Fig. 3e, f). Therefore, the data further support the presence of additional driving force that promotes MAZ LLPS in cells.

### G4 promotes molecular motility in MAZ-containing phase-separated condensates

We next tested whether MAZ/G4 interaction was required for MAZ phase separation. CCND1-G4 WT, Mut1 and Mut2 oligos could

incorporate into pre-formed mCherry-MAZ droplets. However, WT and Mut2, which formed G4s, significantly enhanced MAZ phase separation, but Mut1 showed severely compromised ability, based on droplet numbers and sizes (Fig. 4a). Among them, WT exhibited strongest effects and facilitated MAZ condensation in a concentration-dependent manner (Fig. 4a and Supplementary Fig. 9i). MAZ/CCND1-G4 condensates were also sensitive to temperature, 1,6-hexanediol and salt concentration changes (Supplementary Fig. 10a–d). Strikingly, instead of mCherry-MAZ's gel-like condensates, MAZ/CCND1-G4

**Fig. 3 | MAZ undergoes phase separation in vitro and in cells. a** MAZ protein profiles in cancer tissues. IF staining images of MAZ protein in liver cancer samples and matched normal liver tissues are presented (n = 3 biologically independent experiments). Four pairs of matched samples are presented out of 8 pairs that showed similar results. **b** MAZ protein domain structure and intrinsic disorder analysis using the VSL2 algorithm (left) and its crystal structure (right). Regions with scores > 0.5 indicate intrinsically disordered regions (IDRs). **c** Representative fluorescence and differential interference contrast (DIC) images of mCherry-MAZ droplets at different concentrations in a buffer containing 125 mM NaCl and 10% PEG 8000. The same condition was used hereafter, if not specifically noted. **d** Effects of 1,6-hexanediol (1,6-hex) on mCherry-MAZ punctum formation. HeLa cells expressing mCherry-MAZ were treated with or without 5% 1,6-hex for 2 min and imaged. Zoom: 5× magnification. Scale bar: 2 μm. Quantification of punctum

numbers is presented at right. Data are mean ± s.e.m. (n = 30 cells/each). **e, j** Fusion of adjacent droplets formed by mCherry-MAZ I and mCherry-MAZ$^{ZF3-5}$ (**j**) over time. Representative of three biologically independent experiments. **f, k** FRAP analysis of droplets formed by mCherry-MAZ (**f**) and mCherry-MAZ$^{ZF3-5}$ (**k**). **g, h** Fusion of adjacent puncta formed by mCherry-MAZ over time (**g**), and FRAP analysis of mCherry-MAZ puncta (**h**) in HeLa cells. **i** In vitro droplet formation of mCherry-MAZ WT and mutants shown in Fig. 2e. In (**c, i**), droplets' number and area quantification are shown at right. Data are mean ± s.e.m. (n = 6 fields/each). In (**f, h, k**), FRAP curves for mCherry-MAZ droplets and puncta are shown at right. Data are mean ± s.d. (n = 6 droplets or puncta/each). Percentages of recovery relative to initial fluorescence before photobleaching are provided (**h, k**). Scale bars are either 2 μm in (**e–h, j, k**) or 20 μm in (**c, i**). In (**d, i**), unpaired two-tailed Student's t-test was used for statistical analysis. Source data are provided as a Source Data file.

displayed quick droplet fusion and rapid FRAP (Fig. 4b, c and Supplementary Movie 4). G4 and MAZ signals within their droplets exhibited comparable FRAP rates (Fig. 4c). Importantly, both WT and Mut2 oligos strongly boosted mCherry-MAZ's FRAP versus Mut1 and no oligo (Fig. 4d), indicating that G4s could promote molecular motility of MAZ condensates. This G4's ability was steadily recapitulated in cells, in which mCherry-MAZ exhibited decent FRAP when CCND1-G4 WT oligo was cotransfected versus Mut1, while Mut2's effects fell between WT and Mut1 (Fig. 4e). MAZ binds both dsDNA and G4s, so we tested how the two types of DNA affected MAZ condensation. Both ds-CCND1 (containing CCND1's MAZ binding site) and CCND1-G4 incorporated into mCherry-MAZ droplets, and promoted its phase separation in a dose-dependent manner, but G4s exhibited markedly stronger ability (Supplementary Fig. 10e). Surprisingly, although both types of DNA promoted mCherry-MAZ's FRAP, ds-CCND1 displayed significantly reduced recovery rate versus CCND1-G4, and its mutant form ds-CCND1-M did not integrate into the droplets (Fig. 4f). Furthermore, mCherry-MAZ's droplet formation and FRAP ability could be stimulated by all tested G4-containing oligos of MYC, BCL2, MDM2, MYB and TEL, but not a control poly-adenine (poly A) oligo (Fig. 4g, h). Thus, G4-promoted molecular motility in phase-separated MAZ condensates is likely a general phenomenon in the nucleus.

Next, we evaluated whether G4s regulated MAZ condensation. TMPyP4 or PDS, which disrupted MAZ-G4 interaction (Fig. 2b), significantly reduced MAZ/CCND1-G4 droplet formation (Supplementary Fig. 10f), and also eliminated mCherry-MAZ nuclear puncta in cells (Fig. 4i). The data clearly imply that G4 promotes MAZ phase separation both in vitro and in cells.

Previous studies indicated that G-quartet number and loop length determined G4 structure stability[45]. To examine the effects of these factors on MAZ phase separation, we designed G4-forming oligos containing varying quartet numbers and loop lengths (Supplementary Table 4). In vitro droplet formation experiments indicated that the loop lengths among G-quartets positively correlated with mCherry-MAZ phase separation; with the same loop length, oligos containing 3 G runs showed better stimulative effects on MAZ droplet formation than 2 G runs (Supplementary Fig. 11). Thus, G-quartet number and loop length are proportional to oligos' ability to promote MAZ phase separation.

### Dissect the mechanism underlying MAZ/CCND1-G4 condensation

To interrogate whether G4-promoted MAZ phase separation depended on their direct interaction, we first tested how CCND1-G4 oligo affected droplet formation of MAZ cysteine mutants that either completely or partially lost G4-binding (Fig. 2g). mCherry-MAZ ZF2M, ZF3M and ZF4M, but not ZF234M, still formed droplets with comparable numbers to WT MAZ, despite smaller sizes (Fig. 5a). However, only WT mCherry-MAZ, but none of the mutants, showed robustly enhanced droplet formation in response to CCND1-G4 oligo (Fig. 5b).

In cells, mCherry-MAZ-ZF3M and -ZF4M that retained G4-binding ability formed nuclear puncta, similar to WT MAZ (Fig. 5c and Supplementary Fig. 12a, b). In HL-7702 and HeLa cells expressing low MAZ (Supplementary Fig. 12c), mCherry-ZF2M and -ZF234M, deficient in G4-binding, did not form any visible punctum, despite their diffusive nuclear distribution; however, in HepG2 and SMMC-7721 cells with high endogenous MAZ, these two mutants could still be detected as discrete nuclear puncta (Fig. 5c and Supplementary Fig. 12b, c). Together, G4-binding incapable MAZ mutants, primarily due to defective ZF2, did not autonomously undergo phase separation in cells, but they could still be integrated into endogenous MAZ's puncta. The data also reinforced our hypothesis that G4-binding affinity is indispensable for nuclear MAZ condensation, consistent with observations in Figs. 4i and 5b.

MAZ can both bind G4 and undergo phase separation. To understand how these two functions contributed to G4-promoted MAZ condensation, we premixed EGFP-MAZ and CCND1-G4 in vitro to let them form condensates, and then added mCherry-MAZ WT or each of ZF2M, ΔZF5 and ZF2M-ΔZF5 mutants (Supplementary Fig. 7c, d) that were deficient in G4-binding, phase-separation, or both, respectively. Compared to WT, ZF2M mutant was still incorporated into the green droplets but with reduced efficiency, while ΔZF5 and ZF2M-ΔZF5 virtually lost this ability (Supplementary Fig. 12d, merged images quantified in Supplementary Fig. 12e). The in vitro observation was corroborated by cell-based studies. In HL-7702 cells with low endogenous MAZ, only mCherry-MAZ WT, but none of the mutants, formed nuclear puncta (Supplementary Fig. 12f). In HepG2 and SMMC-7721 cells expressing high MAZ, mCherry-MAZ-ZF2M, but not ΔZF5 and ZF2M-ΔZF5, formed nuclear puncta as MAZ WT did, but its staining was largely diffused when endogenous MAZ was silenced by sh-MAZ-3'UTR, an shRNA targeting MAZ mRNA's 3'-UTR (Fig. 5d). Therefore, MAZ-ZF2M lacking G4-binding but retaining phase separation ability could still incorporate into WT MAZ's condensates, whereas MAZ-ΔZF5 that bound G4 but lacked condensation ability did not show this incorporation (Fig. 5d). Together, we propose a model that G4-MAZ binding through ZF2 initiates their co-condensation, while ZF3-5 can integrate additional MAZ into the condensates (Fig. 5e).

### MAZ/CCND1-G4 condensates compartmentalize transcriptionally active components

Since G4 signals colocalized with gene activation markers (Supplementary Fig. 3a–d), we examined MAZ's subcellular localization. In HepG2 and SMMC-7721 cells, both ectopic mCherry-MAZ and endogenous MAZ colocalized with all tested active transcription markers, including BRD4, MED1, CDK9, Pol II S2P/S5P, H3K27ac and H3K4me3, but not the repressive histone marker H3K9me3 (Fig. 6a and Supplementary Fig. 13, 14a, b). Thus, G4-mediated MAZ condensation likely creates nuclear hubs to accommodate transcriptional activation components. We also observed co-condensate droplets of mCherry-MAZ with EGFP-fused IDR domain of BRD4, MED1 or Pol II (Supplementary Fig. 15a). Importantly, their droplet formation was promoted

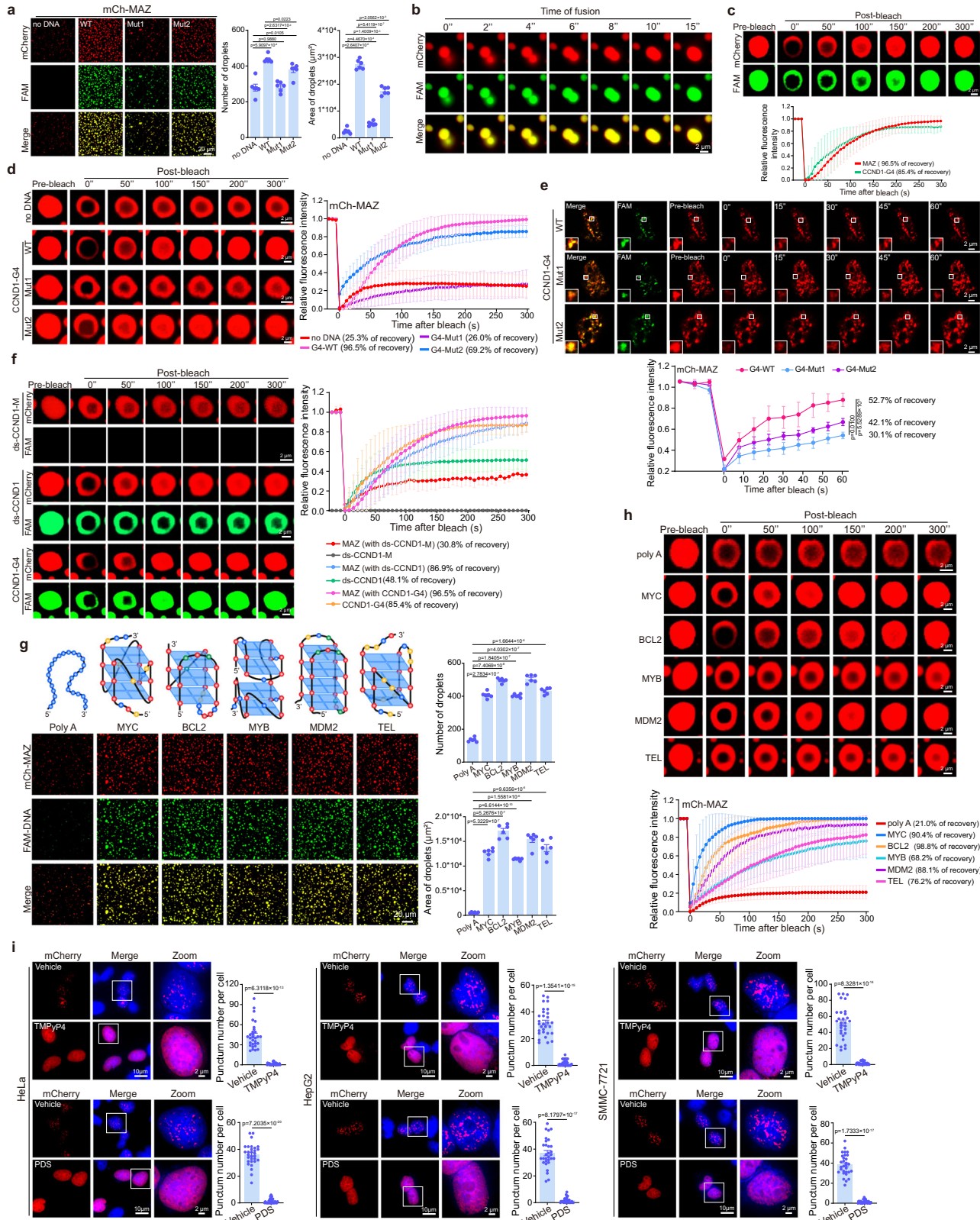

by CCND1-G4 WT oligo, while CCND1-G4 Mut1 showed largely atte-nuated ability (Supplementary Fig. 15b), suggesting that G4s' stimula-tive effect on phase separation is a general phenomenon. Consistently, all these three IDRs could significantly bind CCND1-G4 WT oligo, but not Mut1, in EMSAs (Supplementary Fig. 15c–e). These EGFP-IDRs promptly incorporated into preformed MAZ/CCND1-G4 WT droplets,

but the incorporation markedly dropped when using Mut1 (Fig. 6b). Moreover, in the presence of CCND1-G4 WT, but not Mut1 or absence of oligo, the abovementioned active transcription components or markers were steadily detected in pelleted MAZ droplets after incu-bated with nuclear extracts of HCC cells (Fig. 6c). The enrichment by MAZ/CCND1-G4 WT was literally proportional to increased MAZ

**Fig. 4 | G4s promote MAZ phase-separated condensation. a** Effects of annealed FAM-labelled WT, Mut1 and Mut2 oligos on in vitro mCherry-MAZ droplet formation. **b** Fusion of adjacent droplets formed by MAZ and G4 motifs over time. Representative of three biologically independent experiments. **c** FRAP analysis of MAZ/CCND1-G4 droplets. **d** Effects of annealed FAM-labelled WT, Mut1 and Mut2 oligos on the FRAP of mCherry-MAZ droplets. **e** FRAP analysis of nuclear puncta in HeLa cells cotransfected by mCherry-MAZ and FAM-CCND1-G4 WT, Mut1 or Mut2 oligo. **f** Effects of different types of DNA on FRAP of mCherry-MAZ droplets. CCND1-G4 oligo and its cognate ds-DNA (ds-CCND1) or MAZ binding site-mutated ds-DNA (ds-CCND1-M) were individually mixed with mCherry-MAZ in FRAP assays. **g, h** Effects of different G4s on mCherry-MAZ droplet formation (**g**) and FRAP (**h**). In top of (**g**), different types of G4s, including MYC-G4 (3 G-quartets, parallel), BCL2-G4 (3 G-quartets and long loop, hybrid), MYB-G4 (2 G4s with two G-quartets),

MDM2-G4 (4 G-quartets, antiparallel) and TEL-G4 (3 G-quartets, hybrid) in their corresponding genes or promoters, were schematically presented. **i** Effects of TMPyP4 and PDS on mCherry-MAZ punctum formation. HeLa (left), HepG2 (middle) and SMMC-7721 (right) cells expressing mCherry-MAZ were treated TMPyP4, PDS or vehicle for 24 h, and imaged. Right, quantification of punctum numbers. Data are mean ± s.e.m. ($n = 30$ cells/each). Zoom: 5× magnification. Scale bar, 2 μm. In (**a, g**), droplet quantifications are shown at right. Data are mean ± s.e.m. ($n = 6$ fields each). In (**c–f, h**), quantitative FRAP curves for the droplets or puncta are shown in bottom (**c, e, h**) or right (**d, f**) panels. Data are mean ± s.d. ($n = 6$ droplets each). Percentages of recovery relative to initial fluorescence before photobleaching are provided. Scale bar: 20 μm (**a, g**); 2 μm (**b–f, h**). In (**a, e, g, i**), unpaired two-tailed Student's $t$-test was used for statistical analysis. Source data are provided as a Source Data file.

concentrations (Fig. 6d). Thus, CCND1-G4 promoted MAZ interaction or co-condensation with coactivators, which may subsequently activate target gene expression. Additionally, knockdown of either BRD4 or MED1 significantly reduced endogenous CCND1 expression, suggesting that both coactivators are necessary for CCND1 expression (Supplementary Fig. 16).

### MAZ/CCND1-G4 condensates activate CCND1 transcription that promote cell proliferation and tumor growth

To determine how MAZ/CCND1-G4 condensates contribute to MAZ-mediated transcription and HCC progression, we tested a series of MAZ mutants (Fig. 7a) and purified their mCherry fusion protein (Supplementary Fig. 7c, d). MAZ fusion proteins ΔZF5-FUS-IDR and ZF2M-ΔZF5-FUS-IDR contained FUS protein's IDR (FUS-IDR) at C-termini, which may restore their phase separation ability, as previously reported[41,46]. Compared to WT MAZ, ZF2M showed markedly reduced droplet formation, while ΔZF5 and ZF2M-ΔZF5 lacked this ability, which could be largely restored by FUS-IDR (Fig. 7b). In HeLa cells, WT and the two FUS-IDR fusion proteins formed nuclear puncta, but ZF2M, ΔZF5 and ZF2M-ΔZF5 mutants exhibited diffused signal (Fig. 7c), suggesting that both G4-binding and phase separation abilities are required for MAZ punctum formation in cells with low endogenous MAZ. In reporter assay, ZF2M, ZF2M-ΔZF5 and ZF2M-ΔZF5-FUS-IDR mutants virtually failed to activate the CCND1 promoter, but ΔZF5 retained about half of WT's activity (Fig. 7d), indicating the crucial role of intact ZF2 in MAZ-mediated transcription. Additionally, ΔZF5-FUS-IDR exhibited almost doubled activity of WT. Together, G4 binding ability of ZF2 is critical for MAZ to drive CCND1 expression, which can be greatly promoted by MAZ phase separation.

In HL-7702 cells expressing low MAZ, only ΔZF5-FUS-IDR showed comparable activity in promoting cell proliferation and endogenous CCND1 expression, while most other mutants behaved similarly to negative control (Fig. 7e). MAZ-ΔZF5 could still increase CCND1 levels, but to a substantially reduced extent (Fig. 7e, bottom), recapitulating the observation in reporter assays (Fig. 7d). These mutants were also expressed in HepG2 and SMMC-7721 cells with simultaneously depleted endogenous MAZ by sh-MAZ-3′UTR. Cells expressing ZF2M-ΔZF5 or ZF2M-ΔZF5-FUS-IDR displayed similar growth to negative control, and both ZF2M and ΔZF5 mutants retained low proliferative activity compared to MAZ-WT and empty vector (Fig. 7f, g, top). ΔZF5-FUS-IDR showed similar activities to WT, even higher than that of a control shRNA (sh-Cont) (Fig. 7f, g, top), indicating that further increase of functional MAZ improved cell viability. Additionally, CCND1 was markedly activated by both MAZ-WT and ΔZF5-FUS-IDR, but not other mutants (Fig. 7f, g, bottom). MAZ protein expression or knockdown was verified (Supplementary Fig. 17). Together, our data unequivocally indicated that both G4-binding and phase separation ability are prerequisite for adequate MAZ-promoted CCND1 expression and HCC cell proliferation.

Next, we tested these MAZ mutants in a mouse xenograft model using HepG2 cells with concurrent endogenous MAZ knockdown.

Generally consistent with in vitro data, only MAZ-WT and ΔZF5-FUS-IDR could promote tumor growth, while all other mutants virtually lost this ability, showing tumor growth comparable to vector control (Fig. 7h, i). Meanwhile, only tumors expressing MAZ-WT and ΔZF5-FUS-IDR showed both highly expressed CCND1 (Fig. 7j) and increased Ki-67 levels versus other groups (Fig. 7k).

Overall, we demonstrated a regulatory mechanism that CCND1-G4 recruits MAZ and promotes molecular motility in MAZ condensates, which compartmentalize coactivators to activate CCND1 gene expression, and subsequently augment HCC cell proliferation and tumor progression (Fig. 7l).

## Discussion

Since the discovery of G4s in 1988[47], accumulating evidence has revealed that the structures are formed in living cells under physiological and pathological conditions, and modulate various biological processes[6,10,11,48]. In the current study, we demonstrate G4-recruited MAZ to enhance CCND1 gene expression. Importantly, when validating MAZ's phase separation, we detected its droplets' gel-like properties in vitro characterized by slow droplet fusion and loss of FRAP (Fig. 3e, f). On these aspects, mCherry-MAZ expressed in cells showed much better performance (Fig. 3g, h), despite relatively slow fusion and FRAP of nuclear puncta, compared to previously reported phase-separated TFs[41,49,50]. We also observed enhanced MAZ condensation in all 8 tested liver cancer samples versus matched normal liver tissues (Fig. 3a), suggesting clinical relevance of MAZ phase separation. Clearly, the pathological activity of MAZ condensation requires the analysis of additional clinical samples.

Strikingly, the CCND1-G4 oligos could promote molecular mobility of MAZ condensates both in vitro and in cells. Our data indicate that MAZ binds CCND1-G4s to form G4-MAZ co-condensates that serve as hubs to compartmentalize transcription coregulators, including BRD4, MED1, CDK9 and active Pol II, and subsequently activate CCND1 gene transcription. Importantly, CCND1-G4 oligo promoted the incorporation or co-condensation of MAZ with coactivators (Fig. 6c, d), suggesting that liquid-like property of the condensates favors the accommodation of transcription regulatory proteins.

Additionally, the phenomenon of G4-promoted MAZ condensate motility could be extended to other well-characterized G4s, including MYC, BCL2, MYB, MDM2 and TEL (Fig. 4h). Based on these observations, MAZ potentially regulates a variety of G4-containing oncogenes and even telomere homeostasis. Moreover, CCND1-G4 oligo, but not its Mut1, could also facilitate droplet formation of IDRs from BRD4, MED1 and Pol II (Supplementary Fig. 15b), and bind these IDRs in EMSA studies (Supplementary Fig. 15c–e). Together, our findings strongly support a general regulatory mechanism that promoter G4s may tend to adhere disordered regions or promiscuously blend into the condensates composed by different components of transcription machinery. Subsequently, G4s propel molecular dynamic of compartmentalized TFs and coactivators for gene activation.

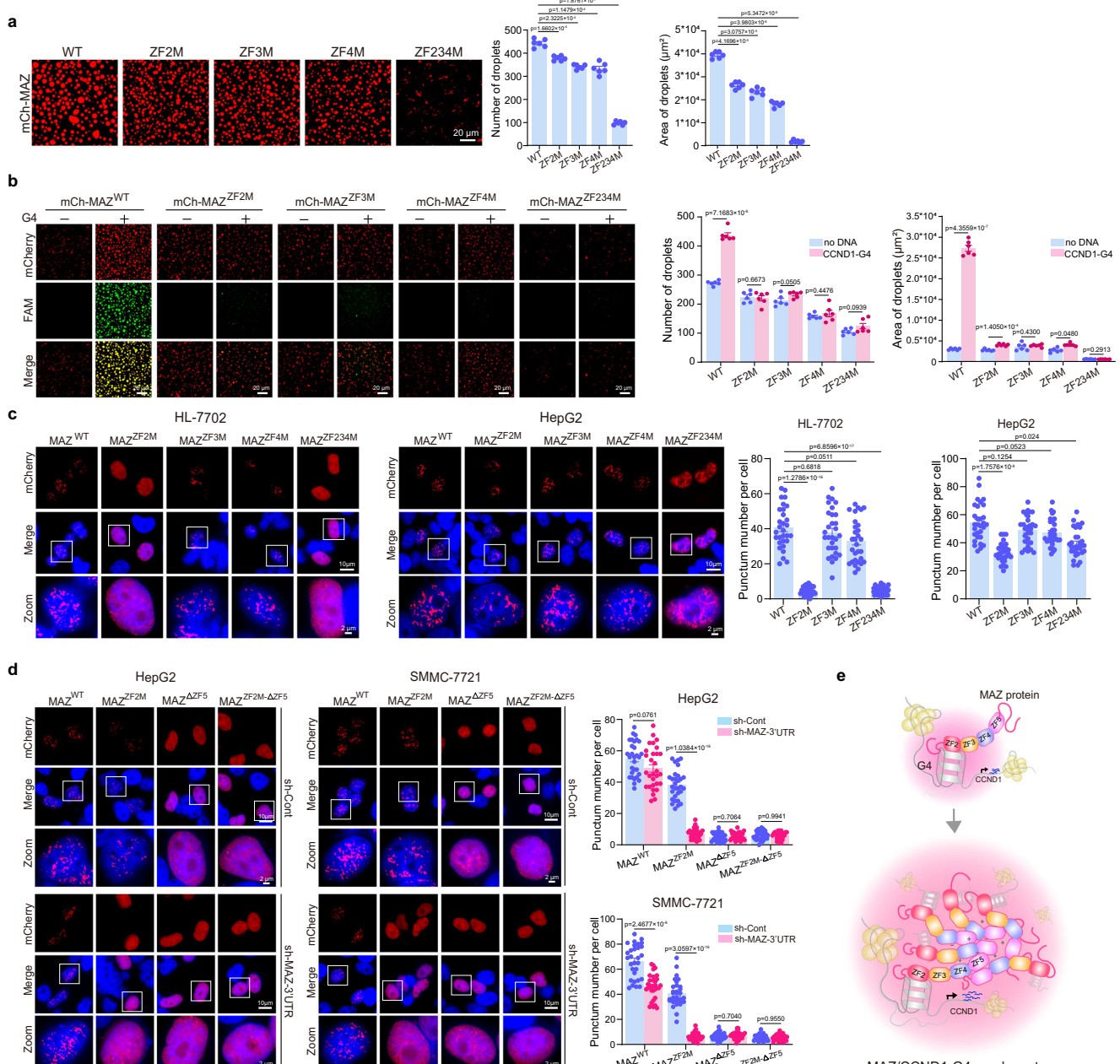

**Fig. 5 | Mechanistic investigation of MAZ/G4 condensate formation. a** In vitro droplet formation assays of mCherry-MAZ WT and indicated mutants. **b** Effects of CCND1-G4 motifs on the droplet formation of mCherry-MAZ WT and its mutants. **c** Detection of punctum formation in HL-7702 and HepG2 cells transfected by expression vectors of MAZ WT and indicated mutants. **d** Detection of punctum formation in HepG2 and SMMC-7721 cells that were infected by lentivirus carrying sh-Cont or sh-MAZ-3′UTR, and transfected by indicated MAZ WT and mutant constructs. **e** A schematic model of G4-recruited MAZ to the CCND1 promoter to induce its phase separation. G4-ZF2 interaction promotes MAZ phase separation,

while the ZF3-5 are responsible for recruiting additional MAZ proteins and coregulators into the phase-separated condensates. In (**a**, **b**), quantification of droplets' numbers and area are shown at right with *P*-values indicated on top. Data are presented as mean ± s.e.m. (*n* = 6 fields each). Scale bar: 20 μm. In (**c**, **d**), quantification of punctum numbers is shown at right with *P*-values indicated on top. Data are presented as mean ± s.e.m. (*n* = 30 cells/each). In each insert, 5× magnification is used and the scale bar is 2 μm. In (**a**–**d**), unpaired two-tailed Student's *t*-test was used for statistical analysis. Source data are provided as a Source Data file.

Interestingly, in FRAP assay, annealed CCND1-G4 oligo, but not its cognate ds-DNA, showed a recovery curve comparable to mCherry-MAZ droplets (Fig. 4f), which was likely attributed to structural difference between G4s and ds-DNAs. G4 is a four-stranded secondary structure consisting of at least two stacked G-quartets[51], and it should more spatially and intimately bind TF than ds-DNA. Additionally, G4 motifs typically show dynamic folding and unfolding conformations[48,52], and different TF molecules may competitively and dynamically bind the same G4[7]. Reciprocally, G4-binding proteins also regulate G4-structural

dynamics[53]. The sophisticated protein-G4 interactions, which are likely absent or different in protein-ds-DNA association, may contribute to both G4s' motile feature in the condensates and their provoking effects on molecular mobility. Clearly, this prediction deserves further investigation.

Many early reports proposed G4s as transcriptional roadblocks to inhibit expression of various proliferative genes, such as MYC, KRAS, MDM2 and YY1[5,54–57]. However, a ChIP-seq study demonstrated the presence of G4 structures in the regions absent of nucleosome and

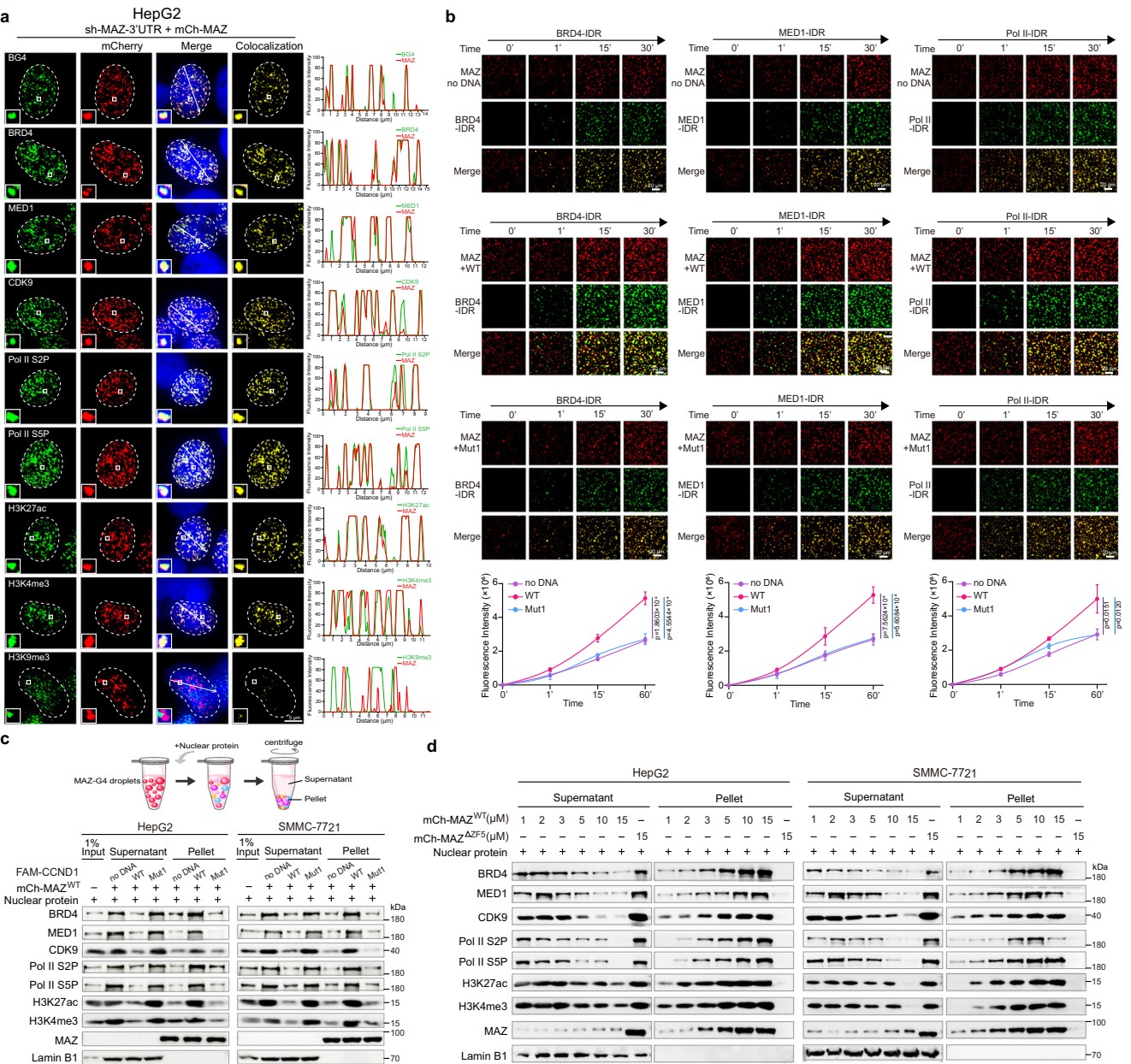

**Fig. 6 | MAZ/CCND1-G4 condensates compartmentalize transcriptionally active components. a** Analysis of mCherry-MAZ colocalization with transcriptional coregulators. HepG2 cells with sh-MAZ-3'UTR-mediated endogenous MAZ knockdown were transfected by mCherry-MAZ plasmid. With mCherry-MAZ in red, cells were stained by antibodies against G4 (BG4), BRD4, MED1, CDK9, active RNA Pol II S2P/S5P, H3K27ac, H3K4me3 and H3K9me3 (in green). Line scans of colocalization images are depicted by white profile arrow lines with quantification shown at right. Scale bar: 5 µm. **b** Time-lapse imaging to examine the effects of CCND1-G4 WT and Mut1 on the incorporation of EGFP-BRD4-IDR, EGFP-MED1-IDR or EGFP-Pol II-IDR (green) into pre-formed mCherry-MAZ/CCND1-G4 droplets (red). Top three panels are samples under conditions of no DNA, WT and Mut1 oligos. Scale bar: 20 µm. Bottom panel is the fluorescence intensity quantification in merged images. Data are mean ± s.d. (n = 3 biologically independent experiments). Unpaired two-tailed

Student's *t*-test was used for statistical analysis. **c, d** Western blot analysis of MAZ/CCND1-G4 droplets-incorporated transcription components from nuclear extracts. In (**c**), purified mCherry-MAZ WT was mixed with annealed oligos of CCND1-G4 WT, Mut1 or equal volume of water, followed by addition of 10 µg nuclear extracts from HepG2 (left panel) or SMMC-7721 (right panel) cells, incubation at 25 °C for 25 min, and centrifugation. Fractionated supernatant and pellet of each sample and 1% of nuclear extract input were analyzed by Western blot using indicated antibodies. In top panel, a schematic diagram simulates the process. In (**d**), mCherry-MAZ WT or ΔZF5 protein mixed with annealed CCND1-G4 WT went through the same procedure as (**c**), followed by Western blot analysis. In (**a, c, d**), data are representative of three biologically independent experiments. Source data are provided as a Source Data file.

close to TSSs, and their association with increased transcriptional activity[10], suggesting a role of G4 motifs in active transcription. Consistently, with recent findings of various G4-binding TFs, increasing evidence unveiled G4s' contribution to gene activation[7,8,33]. Noteworthily, some early studies might have methodological flaws. First, excessively introduced plasmids in reporter assays may not truly reflect G4's regulation on chromatin and even saturate cellular G4-

biding TFs. Second, ligand-bound G4s can exhibit conformations unrecognizable to TFs, as shown in Fig. 2b and Supplementary Fig. 2b of this study. Nevertheless, it is certainly arbitrary to negate these early findings and presume promoter G4s as gene activation markers. G4s likely regulate gene expression in a dynamic manner and the overall outcomes depend on chromatin locations, TF binding, and cellular environment, such as altered potassium levels that showed aberrant

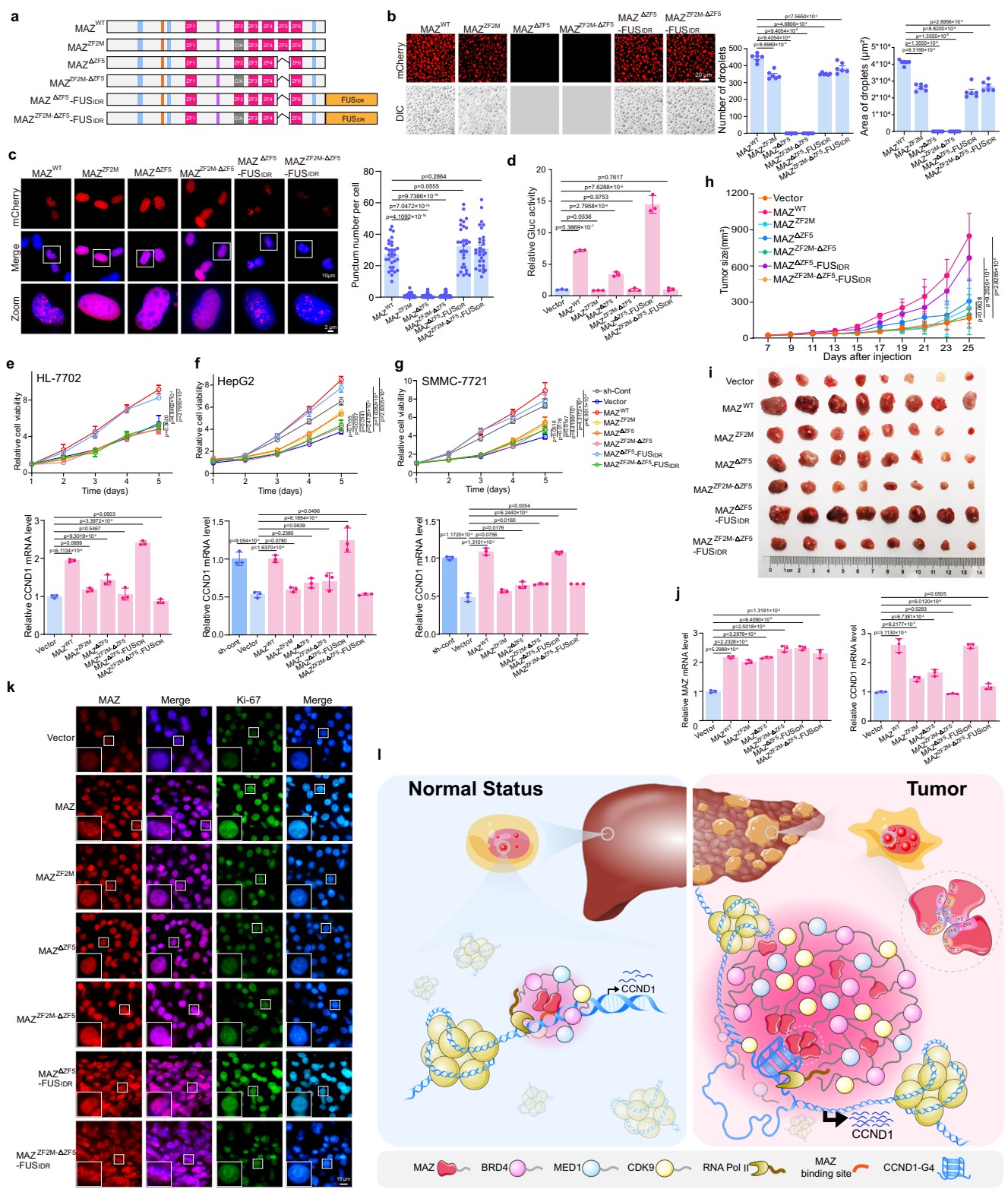

homeostasis in cancer cells[58]. It was reported that transient-to-permanent gene silencing was achieved by converting polycomb complex-mediated H3K27me3 to DNA methylation[59,60]. G4-mediated gene expression may represent more dynamic regulation in response to different physiological and pathological alterations.

CCND1 enhances CDK4 and CDK6 kinase activity to deactivate RB, leading to S phase entry[31]. Consistently, CCND1 overexpression was reported in different cancers[31,61], and its transcription was activated by several proliferative signaling pathways, such as MAPK and PI3K/AKT

pathways[31,62]. In the current study, we discovered a mechanism of CCND1 upregulation, in which promoter G4s recruit MAZ and facilitate its phase separation. Importantly, both condensation capacity and G4-binding affinity were indispensable for MAZ to promote HCC cell proliferation and tumor growth, suggesting the biologically relevance of our discovery.

ZFs has been reported to bind DNA, RNA and protein molecules[63]. In this study, MAZs' ZF could bind G4 motifs and regulate phase separation. Other ZF-containing proteins with G4-binding affinity

**Fig. 7 | Effects of MAZ WT and its mutants on CCND1 expression, HCC cell proliferation and xenograft tumor growth. a** Domain structures of MAZ WT and its phase separation and/or G4-binding deficient or FUS-IDR fusion mutants. **b, c** Droplet formation (**b**) and punctum formation (**c**) of mCherry-MAZ WT and mutants. In (**b**), representative fluorescence and DIC images of droplets (left) and their quantification (right) are presented. Data are mean ± s.e.m. (*n* = 6 fields/each). Scale bar: 20 µm. In (**c**), punctum quantification is shown at right. Data are mean ± s.e.m. (*n* = 30 cells/each). Zoom: 5× magnification. Scale bar: 2 µm. **d** Effects of MAZ WT and mutants on CCND1 promoter activity. Data are mean ± s.d. (*n* = 3 biologically independent experiments). **e–g** Effects of altered MAZ expression on cell viability and CCND1 expression. In (**e**), HL-7702 cells were infected by lentivirus expressing MAZ WT and mutants. In (**f, g**), HepG2 (**f**) and SMMC-7721 (**g**) cells harboring sh-MAZ-3'UTR to silence endogenous MAZ were infected by lentivirus expressing MAZ WT and mutants. Cell viability was determined by proliferation assays (top panel), and CCND1 mRNA levels were examined by RT-qPCR (bottom panel). Data are mean ± s.d. (*n* = 3 biologically independent experiments). **h–k** Effects of MAZ WT and mutants on mouse xenograft tumor growth. HepG2 cells with sh-MAZ-3'UTR-mediated endogenous MAZ knockdown and MAZ WT or mutant expression were grafted into nude mice. Tumor growth curves in about 3 weeks (**h**) and images of excised tumors (**i**) are presented. MAZ and CCND1 mRNA levels (**j**) and IF analyses of Ki-67 expression in xenograft tumors (**k**) are displayed. In (**h**), data are mean ± s.d. (*n* = 8 per group). In (**j**), data are mean ± s.d. (*n* = 3 biologically independent experiments). In (**k**), data are representative of three biologically independent experiments. **l** A schematic model of G4-mediated recruitment of MAZ to form MAZ/CCND1-G4 condensates that compartmentalize coactivators to activate CCND1 gene and subsequently promote hepatocarcinogenesis. In (**b–h, j**), unpaired two-tailed Student's *t*-test was used for statistical analysis. Source data are provided as a Source Data file.

include SP1, PARP1, YY1 and EGR1[9,33,35,64]. Among them, ZFs of YY1 and EGR1 could bind G4 motifs. Previous reports also indicated that multiple ZF proteins from a three-finger library bound telomeric G4s[35], and an engineered Cys2-His2 ZF could recognize G4s[65]. In the current study, we demonstrated that MAZ's ZF2 is responsible for MAZ-G4 interaction (Supplementary Fig. 7e–i).

Many TFs use their IDRs in transactivation domains (TADs) to form phase-separated condensates[14,41]. We also identified high disorder potency in an N-terminal stretch of MAZ. However, this IDR region was dispensable for MAZ phase separation, and did not form droplet when expressed alone. Instead, ZF3-5 with relatively low disorder propensity could steadily form droplets, and their mutants, such as MAZ-ΔZF5, did not undergo phase separation either in vitro or in cells, suggesting that these ZFs are responsible for MAZ phase separation. Similarly, KLF4, a TF containing three C2H2-type ZFs, formed phase-separated condensates depending on its DNA binding, but not predicted IDR[66]; however, whether KLF4's ZFs could stand alone to form condensates was not tested. In addition, PPARγ could undergo phase separation independently of its N-terminal IDR, but relying on its DNA binding domain (DBD) containing a C4-type ZF domain, which could by itself form phase-separated condensates[67]. Interestingly, when tested alone, especially in vitro, both KLF4 and PPARγ condensates exhibited slow fusion and retarded FRAP, similar to MAZ's performance in this study. Whether the gel-like appearance of ZF-mediated protein phase separation is a general phenomenon deserves further investigation.

Collectively, our study discovered a previously unrevealed function of G4s in promoting the molecular motility of MAZ phase-separated condensates through ZF-G4 binding to compartmentalize major coactivators. G4-promoted TF condensate dynamic can activate downstream oncogenes, such as CCND1, and subsequently enhance cancer progression. In recent years, multiple reports revealed that TFs and coactivators form phase-separated condensates on enhancers and promoters, leading to target gene activation[14,30,41]. However, these studies were generally based on TF-binding to the consensus sites on ds-DNA promoters. Our work extended this concept to more spatial regulation, in which structured promoter G4s could recruit these factors and facilitate their phase separation. Importantly, G4s showed better ability to promote protein condensation than ds-DNA, and our findings based on CCND1-G4 and MAZ are unlikely unique but may be extended to G4s in other genes and additional TFs.

## Methods

Our research complies with all relevant ethical regulations and policies by Harbin Medical University Cancer Hospital for de-identified patient samples (permission number HMUIRB2022023) and Northeast Forestry University for xenograft mouse model (permission number 2023074).

### Plasmids, reagents and antibodies

To construct the Gluc expression plasmids in the reporter assays, the amplified CCND1 promoter was ligated into pGLuc-Basic vector to obtain the pCCND1-prmt-Gluc vector (wild-type, WT). The ClonExpress® II Recombination system (Vazyme, C112-01/02) was then used to construct reporter constructs with mutated or deleted G4-forming sequence containing Mut1, Mut2 and del-G4 (Fig. 1g). For the constructs applied in protein expression in bacteria, the coding sequences (CDS) of MAZ (WT and its mutated or truncated mutants) fused with mCherry or EGFP at their N-terminal were inserted into a 6×His expression vector. Bacterial expression vectors for the IDRs of MED1, BRD4 or RNA Pol II were described in our previous report[41]. For the constructs used in gene overexpression or knockdown in cells, the CDS of MAZ (WT or its mutants) were inserted into pmCherry-N1 vector, or a lentiviral vector with a CMV promoter, and a 3×Flag-tag at the N-terminus. Two shRNAs (sh-MAZ-1 and −2) targeting the 3'UTR of the MAZ mRNA, and shCont as a control were generated as described previously[68]. The information about antibodies and their dilution are shown in Supplementary Table 1.

### Circular dichroism (CD) study

The CD assays were performed on a spectropolarimeter (Chirascan, Applied Photophysics Ltd., UK). Oligonucleotides (oligos) shown in Fig. 1a was diluted to 4 µM in 50 mM Tris-HCl buffer supplemented with 50 mM salt (KCl or LiCl). Afterwards, the samples were annealed at 95 °C for 5 min and gradually cooled to 25 °C at a rate of 0.01 °C/s to form G4s. The annealed samples were scanned in a quartz cell of 0.5 mm optical path length to acquire the CD spectra from 200 to 350 nm at 25 °C, and the scan rate was 1 nm/s. For the thermal melting stability of annealed oligos, the CD spectra at 262 nm were recorded every 2 °C between 20 °C and 96 °C with a heating rate of 5 °C/min.

### Electrophoretic mobility shift assay (EMSA)

For protein binding to dsDNA, 0.5 pmol of FAM-labeled dsDNA was incubated with 5 ng of MAZ in a binding buffer containing 250 mM HEPES, 500 mM KCl, 20 mM $MgSO_4$ and 10 mM DTT, pH 8.0. For protein binding to G4, 4 µM of FAM-labelled oligonucleotides were annealed in a 50 mM Tris-HCl buffer added with 50 mM KCl and 40% (w/v) PEG 200. Annealed FAM-labelled oligos were incubated with protein (300 ng of BG4 or 5 ng of MAZ WT, its mutated or truncated mutants) or different amounts of MAZ in the binding buffer above. For the competitive binding assays, the binding reactions were added with unlabeled probes or different amounts of TMPyP4 (Selleck, P1202) or PDS (MedChemExpress, 1085412-37-8). Binding reactions were conducted on ice for 30 min. The samples were then resolved on 8% native gel on ice at 100 V for 50 min (for protein-dsDNA binding), or on 8% native PAGE containing 100 mM KCl and 40% PEG200 on ice at 100 V for 20−24 h (for protein-G4 binding). The gels were scanned on Typhoon FLA7000 (GE, Boston, MA, USA). The probes and cold probes used in the EMSA assays were listed in Supplementary Table 2.

## Liver cancer patient samples

Human liver cancer samples and their adjacent non-tumor samples were obtained from seven male and one female de-identified patients (ages from 57 to 72) at Harbin Medical University Cancer Hospital in China. Written informed consent was signed by all patients. Our study procedure was approved by the Research Ethics Committee of Harbin Medical University (Permission No.: HMUIRB2022023).

## Immunofluorescence staining

Cells were plated on coverslips in 12-well plates and cultured overnight. For immunostaining, cells were fixed in an Immunol Staining Fix Solution (Beyotime, P0098) for 30 min at room temperature, blocked with 10% FBS for 30 min at room temperature, and incubated with a primary antibody for 30 min at room temperature. After washing thrice with PBS, cells were incubated with Alex-Fluor-488- or 594-conjugated secondary antibodies for 1 h at room temperature. Next, cells were washed thrice with PBS, counterstained by DAPI (Beyotime, C1006) to visualize nuclei, and imaged by the GE Delta Vision Elite (GE, Boston, MA, USA).

For IF staining of frozen cancerous samples, sliced sections (10 µm thick/each) were fixed in 4% PFA for 20 min at room temperature, permeabilized with 1% Triton X-100 for 15 min, blocked by 10% goat serum for 1 h at 37 °C, and then incubated with a primary antibody overnight at 4 °C. Subsequently, the sections were washed with PBS thrice, and incubated with Alex-Fluor-488- or 594-conjugated secondary antibodies for 1 h at 37 °C. After washed with PBS, sections were counterstained with DAPI (Beyotime, C1006), and images were acquired using the GE Delta Vision Elite (GE, Boston, MA, USA).

## Chromatin immunoprecipitation (ChIP)

Ten million HCC cells were crosslinked in 1% formaldehyde for 10 min at room temperature, which was quenched with 125 mM glycine. Collected cells were lysed in a nucleic lysis buffer (50 mM Tris-HCl pH 8.1, 10 mM EDTA, 1% SDS and 1× protease inhibitors) to extract nuclei, and then sonicated to an average length of 300–500 bp fragments. Chromatin DNA fragments were incubated with 2 µg BG4 antibody or MAZ antibody, and precipitated using Protein A/G magnetic beads (Thermo Fisher Scientific, 88803) overnight at 4 °C to allow DNA-protein antibody complex formation. The beads were washed with buffer I (0.1% SDS, 1% Triton, 2 mM EDTA, 20 mM Tris-HCl pH 8.1, 150 mM NaCl and 1× protease inhibitors), buffer II (the same as buffer I except for 500 mM NaCl), buffer III (0.25 M LiCl, 1% NP40, 1% Na-deoxycholate, 1 mM EDTA and 10 mM Tris-HCl pH 8.1), and TE buffer (10 mM Tris-HCl pH 8.0, 2 mM EDTA and 1× protease inhibitors). Then, the beads were resuspended with EB buffer (50 mM NaHCO$_3$, 1% SDS), and the crosslinks were reversed at 65 °C for 5 h. DNA fragments were treated by RNase A and proteinase K, extracted by phenol chloroform, and precipitated by ethanol. The immunoprecipitated DNA in all samples were analyzed by gel electrophoresis and qPCR. Primers used in ChIP assays were listed in Supplementary Table 3.

## Dimethyl sulfate footprinting

Four µM of 5′-FAM labelled oligonucleotides (oligos) with or without 100 mM the monovalent cation salt (KCl or LiCl) underwent the annealing procedure as described above. The annealed DNA samples were incubated with 0.2% dimethyl sulfate (DMS, Macklin, D824267) at room temperature for 6 min, quenched by a stop solution (0.1 M β-mercaptoethanol, 0.6 M sodium acetate pH 5.2, 25 mg/ml sonicated salmon sperm DNA and 0.5 mg/ml yeast tRNA), extracted by phenol chloroform, and precipitated by ethanol. Then, the precipitated DNA were resuspended in 10% piperidine, and incubated at 90 °C for 30 min for DNA cleavage. After being air-dried, the DNA samples were resuspended in a solution containing 95% deionized formamide and 5 mM EDTA, and heated at 95 °C for 5 min. Finally, the samples were analyzed by electrophoresis in a 25% denatured polyacrylamide gel at 500 V for

3–4 h, and the gel was imaged using the Typhoon FLA 7000 (GE Healthcare, USA).

## Cell culture, transfection, lentiviral production and infection

HeLa and HEK-293T cells, human hepatic HL-7702 cells, HCC SMMC-7721, HepG2, Hep3B and Huh7 cells were cultured according to the protocols from the American Type Culture Collection (ATCC). HeLa and HEK-293T cells were grown in DMEM medium (Gibco, 11965092) supplemented with 10% fetal bovine serum (FBS) (ExCell Bio, FSP500). HL-7702 cells were cultivated in RPMI 1640 medium (Gibco, 11875093) containing 10% FBS. SMMC-7721 and HepG2 cells were cultivated in MEM medium (Gibco, A1451801) supplemented with 10% FBS. All the cells were incubated at 37 °C in a humidified incubator with 5% CO$_2$ and routinely tested for mycoplasma. The authenticity of cell lines was verified through short tandem repeat (STR) profiling method. Transient transfection assays were performed by Lipofectamine 2000 (Thermo Fisher Scientific, 11668019) according to the protocol provided by the manufacturer. Lentiviral production, infection and cell selection followed our published procedure[69].

## Luciferase reporter assay

For the regulation of G4s on promoter activity, HeLa or HEK-293T cells cultured in 24-well plates were cotransfected with each of reporter vectors (500 ng), and pCMV-SEAP (20 ng) that expressed secreted alkaline phosphatase (SEAP, used as a reference). For the effects of MAZ or its mutants on promoter activity, a reporter vector (250 ng), MAZ expression vector or its mutants (250 ng), and pCMV-SEAP (20 ng) were cotransfected into HeLa cells with or without different amounts of TMPyP4 (Selleck, P1202) or PDS (MedChemExpress, 1085412-37-8). After 48 h of transfection, the Gluc and SEAP activity was measured as described previously by us[54,70].

## Cell proliferation and colony formation assays

For the cell viability assays, cells stably expressing either MAZ shRNAs and/or its cDNAs (WT or its mutants) were split into 96-well plates (3000 cells/well) and cultured overnight. After being cultured at different time points, each well was added with 10 µL of Cell Counting Kit-8 (CCK-8) solution (Bimake, B34304) and incubated at 37 °C for 4 h. Absorbance at 450 nm were measured on a microplate reader (Molecular Devices, LLC., USA).

For colony formation assays, cells stably expressing either MAZ shRNAs or WT MAZ cDNA were split into 6-well plates (10,000 cells/well). After two weeks of culture, colonies in each well were stained using 0.05% crystal violet and then imaged.

## Mouse xenograft study

Animal experiments were approved by the Animal Care and Ethics Committee of Northeast Forestry University. Four-week-old female BALB/c nude mice were purchased from Beijing Weitong Lihua Experimental Animal Technology Co. Ltd. (Beijing, China). Mice were maintained in 12-hour light/dark cycles (6 am–6 pm) at 24 °C with 50–60% humidity and fed standard irradiated rodent chow diet. $4 \times 10^6$ of HepG2 cells carrying MAZ-shRNA-3′UTR with or without MAZ WT or its mutants in 200 µL of 1× PBS/Matrigel (1:1, v:v) (BD Biosciences, 354248) were injected subcutaneously into the right flank of these mice (5-week-old). Tumor sizes were measured by their length (L) and width (W) with a Vernier caliper every three days. Tumor volume (V) was calculated by the following formula: $V = L \times (W^2)/2$. Mice were humanely euthanized at the end of the 3rd week after tumor cell inoculation. The tumor xenografts were removed, and collected for the analysis of immunofluorescence staining and RT-qPCR.

## Western blot analysis

Total proteins of cultured cells or tissues were obtained in the lysis buffer, separated by the SDS-PAGE, and then transferred into PVDF

membranes. The membranes were blocked by 5% nonfat milk at room temperature for 1 h, and subsequently incubated with different primary antibodies at 4 °C overnight and appropriate secondary antibodies at room temperature for 1 h. The immunoreactive signals were visualized by an ECL kit (Vazyme, E411-04/05).

### Reverse transcription and quantitative PCR (RT-qPCR)

Total RNAs were extracted from cultured cells or tissues using the TRIzol reagent (Thermo Fisher Scientific, 15596026). Two µg of the isolated RNAs were reverse-transcribed into cDNA with oligo(dT) primers and M-MLV reverse-transcriptase (Vazyme, R021-01) according to the manufacturer's instructions. The Light-Cycler 480 SYBR Green PCR Master Mix (Roche, S4438) and the primers listed in Supplementary Table 3 were used in qPCR to analyze the cDNA in each group on the Lightcycler 480 instrument (Roche, Basel, Switzerland). All data were normalized against β-actin levels and calculated using the $2^{-\Delta\Delta Ct}$ method.

### Protein expression and purification

His×6-tagged vectors for different proteins were expressed in DE3 *E. coli* cells. Subsequently, bacteria were cultured in LB medium at 37 °C until $OD_{600nm}$ reached 0.6–0.8. Protein expression was induced by 0.2 mM IPTG at 18 °C for 16–20 h. Then, the collected bacteria were lysed in a bacterial lysis buffer (20 mM HEPES, 100 mM KCl, 0.2 mM EDTA, 20% glycerol, 1% Triton, 2 mM PMSF, 1 mM DTT and 1 mg/ml lysozyme, pH 8.0), followed by sonication in ice and centrifuged at 12,000 × g, 4 °C for 20 min. The supernatant containing soluble proteins were incubated with Ni-NTA agarose beads (GE Healthcare, 17057502) for 2 h at 4 °C, and washed with 20 mM imidazole. The bound proteins were eluted by 400 mM imidazole. The purity of eluted proteins was assessed by SDS-PAGE, followed by Coomassie blue staining.

### In vitro phase separation assay

Purified proteins were diluted to different concentrations using 50 mM Tris-HCl (pH 7.4) and 125 mM NaCl. Unless otherwise noted, all in vitro phase separation assays were carried out in a phase separation (PS) buffer containing 125 mM NaCl and 10% PEG 8000 as a crowding agent. For protein phase separation assays, 15 µM proteins or different concentrations of proteins were mixed with the PS buffer with or without different concentrations of NaCl, KCl, or 10% PEGs with indicated molecular weights. For dsDNA/G4-induced phase separation assays, FAM-labelled dsDNA or annealed G4-forming oligonucleotides were added to purified protein mixtures in a modified PS buffer containing 25 mM KCl, 125 mM NaCl and 10% PEG 8000. Five µL of protein or protein/DNA solutions were immediately loaded onto a glass slide, and covered with a coverslip. Fluorescent and differential interference contrast (DIC) microscopy images were taken using the GE Delta Vision Elite with a 60× objective.

Turbidity was measured using BioSpectrometer basic (Eppendorf, Hamburg, Germany). Proteins were treated as described in protein phase separation assays. 60 µL of samples were incubated in Eppendorf tubes for 10 s at ambient temperature, and then the absorbance at $OD_{600nm}$ was measured.

### Fluorescence live-cell imaging

HeLa cells plated on glass-bottom dishes were transfected with mCherry-MAZ expression vector for 24 h. Afterwards, images were acquired every 10 s for 2 h on the live-cell image system (GE Delta Vision Elite) at 37 °C provided with 5% $CO_2$ with a 60× objective.

### Imaging of fluorescence recovery after photo-bleaching (FRAP)

FRAP was conducted on the FRAP module of a Zeiss LSM880 microscope (Zeiss, Germany) using a 63× oil objective, and images were collected using the ZEN software. For FRAP of mCherry-fusion protein

droplets with or without FAM-labelled oligonucleotides, after a circular region of interest (ROI) was selected and three baseline frames were acquired, the ROI was bleached with a 565 nm laser or 488/565 nm Argon laser at a 100% power, and imaged every 2 s for 400 s post-bleaching for fluorescence recovery.

For FRAP of puncta formed by mCherry-MAZ in HeLa cells that were cultured in glass-bottom dishes (NEST, 801001), the selected ROI within a punctum was bleached at a 100% power (565 nm), and imaged every 0.8 s for 20 s post-bleaching for fluorescence recovery. Analyses of the fluorescence intensity of the background region, reference region and bleached region were performed using the FRAP module in the ZEN software.

### HCC cell nuclear extract preparation and droplet pelleting

Cultured HCC cells were washed using ice-cold 1× PBS, and then scraped and centrifuged at 900 × g for 3 min at 4 °C. The cell pellets were lysed in Buffer A (20 mM HEPES, 10 mM NaCl, 3 mM $MgCl_2$, 0.1% NP40, 10% glycerol, 0.2 mM EDTA, 1 mM DTT and 0.4 mM PMSF) on ice for 10–15 min, followed by centrifugating at 400 × g for 5 min at 4 °C. Nuclei in pellets were washed in Buffer B (20 mM HEPES, 10% glycerol, 0.2 mM EDTA, 1 mM DTT and 0.4 mM PMSF), and lysed in Buffer C (20 mM HEPES, 400 mM NaCl, 10% glycerol, 0.2 mM EDTA, 1 mM DTT and 0.4 mM PMSF) on ice for 45 min. After centrifuging at 16,000 × g for 15 min at 4 °C, the supernatants containing the nuclear extract were collected.

For droplet pelleting assays, annealed CCND1-G4 (WT) oligo was incubated with different concentrations of purified mCherry-MAZ protein in PS buffer to form MAZ/CCND1-G4 droplets. Afterwards, the MAZ/CCND1-G4 droplets were mixed with 10 µg of HCC cell nuclear extract, and incubated for 20 min at room temperature. Finally, the samples were centrifuged at 13,000 × g for 10 min, and collected supernatants and pellets were examined by Western blot analysis using the different antibodies.

### Statistical analysis

For each experiment, data were derived from typically three or more biologically independent experiments with similar results. Results were presented as the mean ± standard error of mean (s.e.m.) or standard deviation (s.d.), and sample numbers are described in respective figure legends unless otherwise stated. Unpaired two-tailed Student's *t*-test was conducted for statistical analysis using the software Graphpad Prism (ver.9) or Microsoft Excel (ver.2311). Significant difference is $p < 0.05$, and no significant difference is $p > 0.05$.

### Reporting summary

Further information on research design is available in the Nature Portfolio Reporting Summary linked to this article.

## Data availability

The previously published BG4 ChIP-seq data reused in this study are available in the GEO database (https://www.ncbi.nlm.nih.gov/geo/) under accession number GSM4474689. The other ChIP-seq data reused are available in ENCODE database (https://www.encodeproject.org/) under accession numbers ENCFF094ETW (BRD4), ENCFF136GJN (MED1), ENCFF206GNI (Pol II S2P), ENCFF614QZS (Pol II S5P), ENCFF401PIF (H3K27ac), ENCFF981CDI (H3K4me3), ENCFF290MPO (H3K9me3) and ENCFF873NBY (MAZ). Intrinsically disordered regions of MAZ were analyzed using PONDR database (http://www.pondr.com/). The crystal structural information for MAZ protein (ID: P56270) was obtained from the UniProtKB database (https://www.uniprot.org/). The MAZ-binding sites in the CCND1 promoter were predicted using the Jaspar Matrix database (https://jaspar.elixir.no/). The raw data generated in this study are provided in a Source Data file. Source data are provided with this paper.

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

## Acknowledgements

This work was supported by the National Natural Science Foundation of China (81802798) to D.L. and (82273107) to G.S., the Fundamental Research Funds for the Central Universities (2572022DQ06) to G.S. and (2572021BD03) to D.L., the National Natural Science Foundation of Heilongjiang, China (LH2020H001) to D.L., the China Postdoctoral Science Foundation (2018M631897) and the Heilongjiang Postdoctoral Fund (LBH-Z16005) to D.L.

## Author contributions

W.W., D.L., and G.S. initiated the project, designed the experiments, analyzed data, and wrote the manuscript. W.W., D.L., Q.X., J.C., G.L., S.Q., J.P., and H.W. carried out the research. W.W. and D.L. analyzed the ChIP-seq data. Z.Y. and T.Z. provided essential experimental materials and conceptually contributed to manuscript preparation. J.S. contributed conceptual suggestions. All authors have read and agreed to the published version of the manuscript.

## Competing interests

The authors declare no competing interests.
