## [Peer Review File · Nature Communications]

G-quadruplexes promote the motility in MAZ phase-separated condensates to activate CCND1 expression and contribute to hepatocarcinogenesisREVIEWER COMMENTS

Reviewer #1 (Remarks to the Author):

In the present manuscript, Wang et al found that the G4 motifs of CCND1 gene promoter could recruit transcription factor MAZ to the promoter region and induce LLPS of MAZ, which compartmentalize several transcription coregulators and active Pol II, thus promoting CCND1 gene expression and hepatocarcinogenesis. Actually, the mechanism is an emerging contribution to the molecular understanding of the cancer development. Molecular and biochemical experiments are well-designed and the quality of the data is high. However, the physiological function and significance of this mechanism has not been well described. The authors should validate well the authenticity and importance of this mechanism in clinical samples and in vivo models.

1. Since that G4-promoted phase separation of MAZ is likely a general phenomenon in the nucleus, why does authors focus on CCND1 rather than other G4-containing target genes of MAZ?

2. Recently, G4s were reported to bind to MAZ and the ability of G4s to induce phase separation is well verified. Therefore, the entry of MAZ into G4s-induced condensates is easy to guess, whereas the importance of G4-promoted phase separation of MAZ in tumor malignant progression needs to be further evaluated. Tumor progression is highly complex and dynamic, and it is difficult for xenograft tumor models to simulate cancer development in vivo. Similarly, is this phenomenon present in clinical samples of liver cancer?

3. In fig 2d (right), as mentioned by the authors, the activation of CCND1 reporter by MAZ is dampened by PDS, so why does the Gluc activity contrarily increased under high dose of PDS?

4. The authors mentioned that ZF2 is the most essential element mainly according to fig 2h, in which MAZ(ZF2M) failed to form complex with CCND1-G4 while ZF3M and ZF4M did. However, in fig 2g, there was no difference between mutants containing ZF2 or not, which confused readers of the importance of NF2.

Reviewer #2 (Remarks to the Author):

This manuscript presents a new evidence in G4s-induced phase separation of TFs for gene activation. The authors demonstrate that G-quadruplexes in CCND1 promoter can activate CCND1 expression by promoting MAZ phase separation. They also confirm that the activated oncogene expression leads to exacerbate hepatocarcinogenesis. Overall, the manuscript is clear and the experimental data seems support their claim, but the following comments should be addressed:

1. This manuscript should be concise, both in main text and the Figures. Please make sure the key data are put into Figures and the others could be moved into Supporting Information.

2. The gene activation through phase-separated condensate formation by TFs and their coactivators has been reported (P94). Please give some discussion about the new findings and significance of this work in "discussion" section.

3. In P110, the authors claim "the human CCND1 gene promoter has a high G/C content". That may be the reason for authors to "examine whether CCND1 could be targeted by MAZ (P98-100)", since "the MYC-associated zinc finger (MAZ) protein was previously reported to bind G4s (P89)". The introduction of CCND1 in Result section 1 could be moved into P98 for better understanding.

4. In P173-175, the authors claim that "overlapped enrichment between G4s and active transcription markers, including MED1, BRD4, RNA Pol II, H3K27ac and H3K4me3, in the CCND1 promoter". This result indicated the transcriptional coregulators play a role in CCND1 transcription. If one of coregulators was knocked down, what will happen to the transcription of CCND1? Please provide some experimental data.

5. The authors use mCherry-MAZ for EMSA (P189). It is not necessarily to use mCherry-tagged MAZ for EMSA since mCherry may lead to interference for the binding between MAZ and double-stranded oligos. It is recommended to use His \times 6-MAZ for EMSA.

6. In P287-288, the authors claim that "ds-CCND1 displayed significantly reduced recovery versus

CCND1-G4", please give the explanation.

7. In P359, the authors propose that "all these three IDRs (EGFP-fused IDR domain of BRD4, MED1 or Pol II (Fig. 6b)) could significantly bind CCND1-G4 WT oligo". Does it mean that both the transcription factor (MAZ) and the transcriptional coregulators could bind with G4s? Please give a clear explanation.

8. In P421, "the phenomenon of G4-promoted MAZ condensate motility could be extended to other well-characterized G4s, including MYC, BCL2, MYB, MDM2 and TEL". Please explain the biological importance for the compatibility of G4s-induced phase separation of MAZ.

9. The quality of some EMSA results is not good. The distortion could be found in Figure 1e, Figure 2a and 2b, which undermines the confidence for this work.

Reviewer #3 (Remarks to the Author):

Wang et. al. has systematically explored how G4s in the promoter sequence recruit transcription factors and activate gene expression. They demonstrated an intriguing mechanism of G4 mediated gene expression using an example of CCND1 and MAZ protein. They showed that the G4 in the CCND1 promoter recruited MAZ and promoted their co-condensation, which facilitated the motility of the MAZ condensates. The motility and increased concentration of MAZ and other protein factors in the LLPS droplets activated expression of the CCND1 gene.

This paper described a new transcription regulation mechanism mediated by G4 via LLPS process. The most important observation is that MAZ-WT and MAZ-Mut2 condensates are liquid-like compared to that with Mut1 and no DNA. It is thus reasonable that because of the liquid-like nature of this condensates, more transcription factors can be recruited to activate the gene expression. A series of experiments with different DNAs confirmed that this liquid-like property is due to the G-quadruplex. The design of the experiments is logical, the data presented are clear. Overall, it is a well-thought and well written manuscript describing a new G4 mechanism for gene expression. I support the publication of this paper in Nature Communications. The following are some minor suggestions.

The authors could do a better job by rearranging the figures between SI and main text, for example: SI Fig 9f can be put in Fig.4, which also shows that G4 may be responsible for liquid-like property of MAZ condensates.

Also, sometimes authors could've clearly compared the in-vitro and cellular experiment. For example, Fig.4e has only two experiments with WT and Mut1 whereas the same invitro experiment has all four DNAs. The authors need to explain this.

Have the authors tried to see the TF recruitment ability in MAZ-Mut1 or MAZ-no DNA condensates? This experiment might also tell whether it is really the liquid-like property of the condensate that favors the accommodation of TFs inside the condensates.

REVIEWER COMMENTS

Reviewer #1 (Remarks to the Author):

In the present manuscript, Wang et al found that the G4 motifs of CCND1 gene promoter could recruit transcription factor MAZ to the promoter region and induce LLPS of MAZ, which compartmentalize several transcription coregulators and active Pol II, thus promoting CCND1 gene expression and hepatocarcinogenesis. Actually, the mechanism is an emerging contribution to the molecular understanding of the cancer development. Molecular and biochemical experiments are well-designed and the quality of the data is high. However, the physiological function and significance of this mechanism has not been well described. The authors should validate well the authenticity and importance of this mechanism in clinical samples and in vivo models.

Reply: We thank the reviewer for the positive comments on the experimental design and data quality in this manuscript. We also acknowledge that relatively limited data to validate the mechanism using the clinical samples. In this study, (1) we have tested the biological functions of various MAZ mutants deficient in binding G4s and/or undergoing phase separation using a xenograft mouse model. (2) Meanwhile, we also detected MAZ condensate formation in clinical liver cancer samples and observed increased MAZ condensation in tumors versus matched normal samples. Indeed, our research was restricted by the lack of models using clinical samples to evaluate the contribution of G4-promoted MAZ phase separation in cancer progression, which is a general dilemma faced by the researchers in the G4 field.

The reviewer's comments are addressed point-by-point below.

According to the reviewers' suggestion, to highlight the key findings in this study, we rearranged the data presentation. Specifically, (1) we moved substantial data in previous Figures to current Supplementary Figures in the revised manuscript. (2) Only the data in previous Supplementary Fig. 9f to current Fig. 4f, as suggested by the Reviewer.

To help the reviewer locate the data in the revised manuscript, we list the rearranged Figures/Supplementary Figures (previous-to-current) and new data below:

Fig. 1a to Supplementary Fig. 1a
Fig. 1c to Supplementary Fig. 1d
Fig. 1d to Supplementary Fig. 1e
Fig. 1f to Supplementary Fig. 2a
Fig. 2b to Supplementary Fig. 4f
Fig. 3a to Supplementary Fig. 8
Fig. 4h to Supplementary Fig. 10f
Fig. 5d to Supplementary Fig. 12d
Fig. 5e to Supplementary Fig. 12e
Fig. 6b to Supplementary Fig. 15a
Fig. 6c to Supplementary Fig. 15b
Fig. 7e-g Western blot to Supplementary Fig. 17
Supplementary Fig. 9f to Fig. 4f
For Reviewer 2-Q4: new results in Supplementary Fig. 16
For Reviewer 3-Q3: new results in Fig. 6c

1. Since that G4-promoted phase separation of MAZ is likely a general phenomenon in the

nucleus, why do authors focus on CCND1 rather than other G4-containing target genes of MAZ?

Reply: We thank the reviewer for the insightful question. We chose the CCND1 (cyclin D1) as the target gene of MAZ for two reasons. First, it is a well-characterized proliferative gene that we have been working on for years. Second, the presence of G4s in the CCND1 promoter and their role in regulating CCND1 expression have not been reported previously, and thus the inclusion of these data will add novelty to the current study. Meanwhile, we also tested the effects of G4s from other genes on MAZ phase separation to evaluate whether this regulatory mechanism is a general phenomenon. Nevertheless, in the revised manuscript, we added the discussion regarding the potential of MAZ in regulating other G4-containing genes targeted by MAZ (lines 393-395).

2. Recently, G4s were reported to bind to MAZ and the ability of G4s to induce phase separation is well verified. Therefore, the entry of MAZ into G4s-induced condensates is easy to guess, whereas the importance of G4-promoted phase separation of MAZ in tumor malignant progression needs to be further evaluated. Tumor progression is highly complex and dynamic, and it is difficult for xenograft tumor models to simulate cancer development in vivo. Similarly, is this phenomenon present in clinical samples of liver cancer?

Reply: We thank the reviewer for bringing up this important point. Currently, to prove the role of G4 in promoting MAZ phase separation and tumor progression, cell-based and mouse xenograft models are still currently available and feasible approaches to be employed. The reason is that no natural patient-derived MAZ mutation with defective G4-binding affinity has been identified or reported. Nevertheless, we attempted to directly treat patient-derived tumors by G4 ligands and 1,6-hexanediol. However, these clinical samples first needed to be frozen and sliced prior to the treatment by these compounds, followed by immunostaining using MAZ antibody. The freezing process led to cell death that made the cells irresponsive to either G4 ligands or 1,6-hexanediol treatment. Overall, due to technical limitation, direct use of clinical samples to evaluate the contribution of G4s-induced MAZ condensation to tumor progression is still not practicable. Cell-based assays and mouse models are likely the only available methods for this type of studies.

3. In fig 2d (right), as mentioned by the authors, the activation of CCND1 reporter by MAZ is dampened by PDS, so why does the Gluc activity contrarily increased under high dose of PDS?

Reply: We thank the reviewer for the question. The effects of G4 ligands on the expression of G4-containing genes depend on their binding positions on G4 structures and ligand concentrations, among others. However, there is still uncertainty regarding the overall effects, especially for reporter genes due to their high copy numbers and non-chromatin environment after being transfected into cells. TMPyP4 binds to G4s through both groove-bound and top-face bound conformations (PMID: 34989224), while PDS interacts with G4s' grooves and DNA phosphate backbone (PMID: 35749293). The difference between the binding patterns of the two ligands may determine their different effects on reporters, especially at relatively high concentrations. Therefore, we predict that the elevated or reinstated luciferase activity of empty vector and MAZ reporter, respectively, at relatively high concentrations of PDS, but not

TMPyP4 (previous Figure 2d, current Figure 2c), is likely due to PDS's binding affinity to DNA phosphate backbone, which needs to be verified in future studies. Nevertheless, the data in Figure 2c clearly indicated that increased PDS levels could diminish the MAZ reporter activity when compared to the vector control. We added the description for this point in the revised manuscript (lines 187-189).

4. The authors mentioned that ZF2 is the most essential element mainly according to fig 2h, in which MAZ(ZF2M) failed to form complex with CCND1-G4 while ZF3M and ZF4M did. However, in fig 2g, there was no difference between mutants containing ZF2 or not, which confused readers of the importance of NF2.

Reply: We thank the reviewer for the question. In current Figure 2g (previous Figure 2h), we used "full-length" MAZ mutants with C-to-A mutations in ZFs, and the EMSA results showed that ZF2 is essentially required for MAZ to bind CCND1-G4. However, although ZF2 is required for MAZ-binding to G4, it is not sufficient for this binding. Therefore, in Figure 2f (previous Figure 2g) using "truncated" MAZ proteins, MAZ(ZF2-4), but not MAZ(ZF2-3), could bind CCND1-G4. Overall, the data suggested that ZF2 is essential and ZF2-4 is sufficient for MAZ binding to CCND1-G4. In the revised manuscript, we rephrased the description to eliminate the ambiguity (lines 205-206).

Reviewer #2 (Remarks to the Author):

This manuscript presents a new evidence in G4s-induced phase separation of TFs for gene activation. The authors demonstrate that G-quadruplexes in CCND1 promotor can activate CCND1 expression by promoting MAZ phase separation. They also confirm that the activated oncogene expression leads to exacerbate hepatocarcinogenesis. Overall, the manuscript is clear and the experimental data seems support their claim, but the following comments should be addressed:

1. This manuscript should be concise, both in main text and the Figures. Please make sure the key data are put into Figures and the others could be moved into Supporting Information.

Reply: We thank the reviewer for the suggestion. In the revised manuscript, we went through the manuscript and simplified many descriptions to make the them as concise as possible, especially the *Results* section. We have also rearranged some results between Figures and Supplementary Figures of previous submission, to highlight our key findings.

To help the reviewer locate the data in the revised manuscript, we list the rearranged Figures/Supplementary Figures (previous-to-current) and new data below:

- Fig. 1a to Supplementary Fig. 1a
- Fig. 1c to Supplementary Fig. 1d
- Fig. 1d to Supplementary Fig. 1e
- Fig. 1f to Supplementary Fig. 2a
- Fig. 2b to Supplementary Fig. 4f
- Fig. 3a to Supplementary Fig. 8
- Fig. 4h to Supplementary Fig. 10f

Fig. 5d to Supplementary Fig. 12d
Fig. 5e to Supplementary Fig. 12e
Fig. 6b to Supplementary Fig. 15a
Fig. 6c to Supplementary Fig. 15b
Fig. 7e-g Western blot to Supplementary Fig. 17
Supplementary Fig. 9f to Fig. 4f
For Reviewer 2-Q4: new results in Supplementary Fig. 16
For Reviewer 3-Q3: new results in Fig. 6c

2. The gene activation through phase-separated condensate formation by TFs and their coactivators has been reported (P94). Please give some discussion about the new findings and significance of this work in “discussion” section.

Reply: We thank the reviewer for the suggestion. Accordingly, in the revised manuscript, we have added these discussions in lines 460-467 of the Discussion section. This part of the discussion reads as:

In recent years, multiple reports revealed that TFs and coactivators form phase-separated condensates on enhancers and promoters, leading to target gene activation (PMID: 30449618, 29930091 and 35390165). However, these studies were generally based on TF-binding to the consensus sites on ds-DNA promoters. Our work extended this concept to more spatial regulation, in which structured promoter G4s could recruit these factors and facilitate their phase separation. Importantly, G4s showed better ability to promote protein condensation than ds-DNA, and our findings based on CCND1-G4 and MAZ are unlikely unique but may be extended to G4s in other genes and additional TFs.

3. In P110, the authors claim “the human CCND1 gene promoter has a high G/C content”. That may be the reason for authors to “examine whether CCND1 could be targeted by MAZ (P98-100)”, since “the MYC-associated zinc finger (MAZ) protein was previously reported to bind G4s (P89)”. The introduction of CCND1 in Result section 1 could be moved into P98 for better understanding.

Reply: We fully agree with the reviewer on this point and feel this can improve the logic of the introduction. In the revised manuscript, we have made the change of the description as suggested by the reviewer (lines 95-97).

4. In P173-175, the authors claim that “overlapped enrichment between G4s and active transcription markers, including MED1, BRD4, RNA Pol II, H3K27ac and H3K4me3, in the CCND1 promoter”. This result indicated the transcriptional coregulators play a role in CCND1 transcription. If one of coregulators was knocked down, what will happen to the transcription of CCND1? Please provide some experimental data.

Reply: We thank the reviewer for the insightful suggestion. In HepG2 and SMMC-7721 cells with individually knockdown of BRD4 or MED1, we observed downregulation of endogenous CCND1, indicating that both coactivators are required for the optimal CCND1 transcription. The data are presented in Supplementary Figure 16 of the revised manuscript (text in lines 331-

333).

5. The authors use mCherry-MAZ for EMSA (P189). It is not necessarily to use mCherry-tagged MAZ for EMSA since mCherry may lead to interference for the binding between MAZ and double-stranded oligos. It is recommended to use His \times 6-MAZ for EMSA.

Reply: We thank the reviewer for the suggestion. We repeated the EMSA studies using the purified His \times 6-MAZ and observed the similar results as previously tested using mCherry-MAZ. The new data were used to replace the previous data in Supplementary Figure 4c-e.

6. In P287-288, the authors claim that “ds-CCND1 displayed significantly reduced recovery versus CCND1-G4”, please give the explanation.

Reply: We thank the reviewer for the suggestion. First, each G4 is a four-stranded secondary structure containing at least two stacked G-quartets stabilized by Hoogsteen hydrogen bonds and should be more stereoscopic and sophisticated than regular ds-DNA. Therefore, it is reasonable to predict that G4 may have more intimate interaction with MAZ and thus cause greater structural impact on MAZ than that of ds-DNA. Second, many G4 motifs typically show dynamic folding and unfolding conformations (PMID: 24920827), and different TF molecules may competitively and dynamically bind the same G4 (PMID: 33892767). Reciprocally, G4-binding proteins also regulate G4-structural dynamics (PMID: 29470465). Collectively, these unique properties of protein-G4 interactions, which are likely absent or different in protein-ds-DNA association, may contribute to both G4s' motile feature in the condensates and their provoking effects on molecular mobility. Therefore, the differences between the two types of DNA structures may lead to the observation of ds-CCND1 displaying significantly reduced recovery versus CCND1-G4 in FRAP assays. We have added these discussions in the revised manuscript (lines 403-411).

7. In P359, the authors propose that “all these three IDRs (EGFP-fused IDR domain of BRD4, MED1 or Pol II (Fig. 6b)) could significantly bind CCND1-G4 WT oligo”. Does it mean that both the transcription factor (MAZ) and the transcriptional coregulators could bind with G4s? Please give a clear explanation.

Reply: We thank the reviewer for bringing up this important point. We observed that CCND1-G4 WT oligo, but not its Mut1 or “no DNA”, could promote the condensate formation of EGFP-fused IDRs of BRD4, MED1 and Pol II (previous Figure 6c, current Supplementary Figure 15b). Meanwhile, in EMSA studies, these fusion IDRs could individually bind to CCND1-G4 WT, but not Mut 1 (previous Supplementary Figure 14, current Supplementary Figure 15c-e). Collectively, these data indicate that CCND1-G4 can interact with the IDR regions of these coregulators. As for the mechanism underlying this interaction, it is possible that G4s may be sticky to protein segments with disordered properties or can integrate into phase-separated condensates. In the revised manuscript, we rephrased the related discussions to manifest these observations (lines 396-399).

8. In P421, “the phenomenon of G4-promoted MAZ condensate motility could be extended to other well-characterized G4s, including MYC, BCL2, MYB, MDM2 and TEL”. Please explain the biological importance for the compatibility of G4s-induced phase separation of MAZ.

Reply: We thank the reviewer for this important question. Indeed, we detected MAZ binding to the annealed oligonucleotides of CCND1, MYC, BCL2, MYB, MDM2 and TEL G4s (Supplementary Figure 5). Meanwhile, these G4s also promoted the molecular motility of MAZ condensates when tested by FRAP experiments (previous Figure 4g, current Figure 4h). The biological significance of these observations is that G4 structures in promoters can generally recruit MAZ to modulate gene expression, while MAZ may also bind TEL G4s to regulate telomere homeostasis. Whether this mechanism can be applied to additional G4s or other TFs deserves future investigation. In the revised manuscript, we elaborated the biological importance of these observation in the Discussion section (lines 393-395).

9. The quality of some EMSA results is not good. The distortion could be found in Figure 1e, Figure 2a and 2b, which undermines the confidence for this work.

Reply: We thank the reviewer for pointing out this for us. During the revision, we have repeated these EMSA experiments multiple times and obtained constant results that are consistent with our initial observations. The images in current Figure 1b (previous Figure 1e), Figure 2a and Supplementary Figure 4f (previous Figure 2b) have been replaced by these with significantly improved quality.

Reviewer #3 (Remarks to the Author):

Wang et. al. has systematically explored how G4s in the promoter sequence recruit transcription factors and activate gene expression. They demonstrated an intriguing mechanism of G4 mediated gene expression using an example of CCND1 and MAZ protein. They showed that the G4 in the CCND1 promotor recruited MAZ and promoted their co-condensation, which facilitated the motility of the MAZ condensates. The motility and increased concentration of MAZ and other protein factors in the LLPS droplets activated expression of the CCND1 gene.

This paper described a new transcription regulation mechanism mediated by G4 via LLPS process. The most important observation is that MAZ-WT and MAZ-Mut2 condensates are liquid-like compared to that with Mut1 and no DNA. It is thus reasonable that because of the liquid-like nature of this condensates, more transcription factors can be recruited to activate the gene expression. A series of experiments with different DNAs confirmed that this liquid-like property is due to the G-quadruplex. The design of the experiments is logical, the data presented are clear. Overall, it is a well-thought and well written manuscript describing a new G4 mechanism for gene expression. I support the publication of this paper in Nature Communications. The following are some minor suggestions.

The authors could do a better job by rearranging the figures between SI and main text, for example: SI Fig 9f can be put in Fig.4, which also shows that G4 may be responsible for liquid-

like property of MAZ condensates.

Reply: We thank the reviewer for the suggestion. In the revised manuscript, we have made rearrangement of some results between Figures and Supplementary Figures of previous submission, to highlight our key findings. As suggested by the reviewer, Supplementary Figure 9f is now Fig 4f.

To help the reviewer locate the data in the revised manuscript, we list the rearranged Figures/Supplementary Figures (previous-to-current) and new data below:

Fig. 1a to Supplementary Fig. 1a
Fig. 1c to Supplementary Fig. 1d
Fig. 1d to Supplementary Fig. 1e
Fig. 1f to Supplementary Fig. 2a
Fig. 2b to Supplementary Fig. 4f
Fig. 3a to Supplementary Fig. 8
Fig. 4h to Supplementary Fig. 10f
Fig. 5d to Supplementary Fig. 12d
Fig. 5e to Supplementary Fig. 12e
Fig. 6b to Supplementary Fig. 15a
Fig. 6c to Supplementary Fig. 15b
Fig. 7e-g Western blot to Supplementary Fig. 17
Supplementary Fig. 9f to Fig. 4f
For Reviewer 2-Q4: new results in Supplementary Fig. 16
For Reviewer 3-Q3: new results in Fig. 6c

Also, sometimes authors could've clearly compared the in-vitro and cellular experiment. For example, Fig. 4e has only two experiments with WT and Mut1 whereas the same in vitro experiment has all four DNAs. The authors need to explain this.

Reply: We thank the reviewer for the question. When testing the effects of annealed oligos on the FRAP of MAZ in vitro (Figure 4d), we carried out experiments under four conditions: in the presence of WT, Mut1 (no G-tract) or Mut2 (with well-separated G-tracts) oligo, or in the absence of any oligo/DNA, and observed that Mut1's ability in promoting MAZ's FRAP fell between WT and Mut2 oligos.

When conducting the cellular experiments, we actually included the three conditions of WT, Mut1 and Mut2, but not "no DNA" that was already presented in current Figure 3h (previous Figure 3i). We initially did not present the data of Mut2, because its sequence is not naturally present and we felt that the comparison between WT and Mut1 should be sufficient to draw our conclusion. Nevertheless, in the revised manuscript, we incorporated Mut2's data into Figure 4e. As shown in this figure, Mut2 retained stimulative activity on MAZ FRAP, although the effects were lower than WT but greater than Mut1, consistent with the in vitro FRAP results. The description of these results is in lines 251-253).

Have the authors tried to see the TF recruitment ability in MAZ-Mut1 or MAZ-no DNA condensates? This experiment might also tell whether it is really the liquid-like property of the condensate that favors the accommodation of TFs inside the condensates.

Reply: We thank the reviewer for this important suggestion for current Figure 6d (previous Figure 6e) in our previous submission. Accordingly, we carried out the experiment to compare the recruitment of MAZ/CCND1-G4(WT), MAZ/MAZ-Mut1 and MAZ/no DNA. In Figure 6c of the revised manuscript, we presented the results of this experiment, indicating that only CCND1-G4(WT), but not MAZ/MAZ-Mut1 or no oligo DNA, could promote MAZ to recruit or precipitate BRD4, MED1, CDK9 and active Pol II, as well as histones H3K27ac and H3K4me3 modifications (lines 326-328). The data indicated that CCND1-G4 oligo promoted the incorporation or co-condensation of MAZ with coactivators, and the liquid-like property of the condensates favors the accommodation of transcription regulatory proteins, as pointed by the reviewer. We added this important point to the Discussion section of the revised manuscript (line 389-391).

REVIEWERS' COMMENTS

Reviewer #1 (Remarks to the Author):

The authors' response to the previous critiques demonstrates a comprehensive and detailed effort. The revised manuscript has addressed, to varying extents, the requests outlined in my prior comments.

One more point, regarding my concern 1. The fact that the authors have previously studied CCDN1 or the association between G4 and CCDN1 has not been reported does not suffice as a reason. However, the authors have supplemented their response with sufficient experimental data, mitigating potential ambiguities. It seems that authors were lucky enough to find an association of G4-driven MAZ phase separation with CCND1.

In addition, the authors briefly demonstrated that MAZ undergoes phase separation in four tumor samples, and its core molecules and mechanisms are barely addressed in terms of expression changes, correlation analysis, and subcellular distribution in clinical samples. Moreover, the relatively small number of clinical samples tested was not sufficient to support this conclusion.

While the author has incorporated new data in the revised manuscript and organized it in the response letter, I would have preferred a more direct presentation of the relevant data in response to the concerns I raised. The current format makes it challenging to correlate the added data with the author's response.

Reviewer #2 (Remarks to the Author):

The authors have carried out a thorough revision of the manuscript, which improved the quality of this work. I recommend publishing the manuscript as it is.

Reviewer #1 (Remarks to the Author):

The authors' response to the previous critiques demonstrates a comprehensive and detailed effort. The revised manuscript has addressed, to varying extents, the requests outlined in my prior comments.

One more point, regarding my concern 1. The fact that the authors have previously studied CCND1 or the association between G4 and CCND1 has not been reported does not suffice as a reason. However, the authors have supplemented their response with sufficient experimental data, mitigating potential ambiguities. It seems that authors were lucky enough to find an association of G4-driven MAZ phase separation with CCND1.

Reply: We thank the reviewer for the comments. The reason to choose CCND1 for the G4 study was initially attributed to the high G/C content and the presence of multiple G-tracts in the CCND1 promoter, which are the characteristic features for potential G4-structure formation. Additionally, well-studied activity of CCND1 in promoting oncogenesis is another reason for us to focus on this gene. Importantly, in addition to our demonstration of MAZ-regulated CCND1 expression through the G4 motifs in its promoter, we also observed MAZ binding to the G4 motifs derived from the promoters of other genes (MYC, BCL2, MYB and MDM2). We also detected MAZ's co-condensation with the oligos containing these G4 motifs, suggesting potential regulation of these genes by MAZ. Certainly, restricted by the length of the current manuscript, we did not further investigate MAZ-mediated expression of these additional genes and only focused on MAZ-mediated CCND1 expression through binding to its G4 motifs.

In addition, the authors briefly demonstrated that MAZ undergoes phase separation in four tumor samples, and its core molecules and mechanisms are barely addressed in terms of expression changes, correlation analysis, and subcellular distribution in clinical samples. Moreover, the relatively small number of clinical samples tested was not sufficient to support this conclusion.

Reply: We thank the reviewer for the comments. When we initially prepared the manuscript, we totally used 8 clinical liver cancer samples for Western blot, RT-qPCR (data shown below; also presented as Supplementary Figure 8b-d of the revised manuscript) and immunofluorescence (IF) studies (data shown below as Figure 1 of this file; cases 1-4 are presented in Figure 3a, while cases 5-8 are not presented in the manuscript). In all 8 samples, the IF studies showed very similar results. Therefore, we presented the images of 4 samples in Figure 3a of the manuscript, which represented the results of 8 samples. However, prior to the manuscript submission, we wanted to focus on G4-promoted MAZ phase separation studies, and thus deleted the data of Western blot and RT-qPCR (in which we mentioned 8 clinical samples), but we forgot to mention that we examined 8 samples in the IF study. We apologize for this negligence.

In the revised manuscript, we added this information (see lines 209-213 in Fig. 3a legend of the revised manuscript). Certainly, 8 samples do not represent a large cohort of patients, although the data from them could clearly support our results generated from the *in vitro* and cell-based experiments. We have acknowledged this limitation in the revised manuscript.

Supplementary Figure 8b-d. MAZ expression in liver cancer tissues. **b**, Western blot analyses to compare MAZ protein expression between 8 normal liver cancer samples and correspondingly matched para-cancerous normal liver tissues. **c**, **d**, MAZ expression from RT-qPCR analyses of the 8 pairs of liver cancer and the matched para-cancerous normal tissues (**c**), and the analysis of a TCGA Liver Hepatocellular Carcinoma (TCGA-LIHC) dataset consisting of 371 cancer samples and 50 para-carcinoma tissues (**d**).

Figure 1. IF staining images of the endogenous MAZ protein in liver cancer samples and their matched normal liver tissues ($n = 8$). Note: Images of cases 1-4 are presented in Figure 3a of the revised manuscript.

For the other comments of the reviewer:

(1) In most studies, mechanistic investigations of protein phase separation and G4-promoted condensation were carried out using *in vitro* and cell-based experiments, and no reliable or well-established approach for this type of studies is available using clinical samples. In our attempts, we found that cells in fixed clinical liver cancer samples did not show molecular motility.

(2) Regarding MAZ levels in cancers, one previous report indicated MAZ overexpression in liver cancers; thus, we did not present our relevant data in our manuscript. Actually, we already analyzed MAZ expression in liver cancer cell lines and the 8 clinical samples, and also explored transcriptomic data of liver cancer samples. Since the reviewer inquired about MAZ expression changes, we added these data in Supplementary Figure 8 of the revised manuscript.

(3) As for the correlation of MAZ phase separation between liver cancer samples and normal liver tissues, we analyzed 8 clinical samples with similar observation that MAZ condensation was very significant in liver cancer samples versus matched normal liver tissues (Figure 3a of the revised manuscript). Certainly, examination of a large number of samples will further verify this conclusion. We acknowledge this limitation in our data in the revised manuscript (see lines 390-393 in the Discussion section).

(The added paragraphs read as: We also observed enhanced MAZ condensation in all 8 tested liver cancer samples versus matched normal liver tissues (Fig. 3a), suggesting clinical relevance of MAZ phase separation. Clearly, the pathological activity of MAZ condensation requires the analysis of additional clinical samples.)

(4) Regarding the subcellular localization of MAZ and its condensate puncta, we observed their nuclear presentation, which is in line with the transcription factor identity of MAZ. Again, studies of more samples will certainly reinforce this conclusion.

While the author has incorporated new data in the revised manuscript and organized it in the response letter, I would have preferred a more direct presentation of the relevant data in response to the concerns I raised. The current format makes it challenging to correlate the added data with the author's response.

Reply: We thank the reviewer for the comment. We apologize for the inconvenient presentation format in our response letter, which made it difficult for the reviewer to correlate the revised content with our response.